# Water in peripheral TM-interfaces of Orai1-channels triggers pore opening
Valentina Hopl[1,5], Adéla Tiffner[1,2,5], Armin Wutscher[1], Matthias Sallinger [1], Herwig Grabmayr [1], Magdalena Prantl[1], Maximilian Fröhlich[1], Julia Söllner[1], Sarah Weiß[1], Hadil Najjar[1], Yuliia Nazarenko[1], Selina Harant[1], Natalia Kriško[1], Marc Fahrner [1], Christina Humer[1], Carmen Höglinger[1], Heinrich Krobath [3], Daniel Bonhenry [4]✉ & Isabella Derler [1]✉

The activation of the $Ca^{2+}$-channel Orai1 via the physiological activator stromal interaction molecule 1 (STIM1) requires structural rearrangements within the entire channel complex involving a series of gating checkpoints. Focusing on the gating mechanism operating along the peripheral transmembrane domain (TM) 3/TM4-interface, we report here that some charged substitutions close to the center of TM3 or TM4 lead to constitutively active Orai1 variants triggering nuclear factor of activated T-cell (NFAT) translocation into the nucleus. Molecular dynamics simulations unveil that this gain-of-function correlates with enhanced hydration at peripheral TM-interfaces, leading to increased local structural flexibility of the channel periphery and global conformational changes permitting pore opening. Our findings indicate that efficient dehydration of the peripheral TM-interfaces driven by the hydrophobic effect is critical for maintaining the closed state of Orai1. We conclude that a charge close to the center of TM3 or TM4 facilitates concomitant hydration and widening of peripheral TM interfaces to trigger constitutive Orai1 pore opening to a level comparable to or exceeding that of native activated Orai1.

The dynamics of calcium ($Ca^{2+}$) control vital cellular processes and functions[1–3]. A well-known $Ca^{2+}$ entry pathway into the cell is the ubiquitously expressed $Ca^{2+}$ release-activated $Ca^{2+}$ (CRAC) channel which is responsible for various physiological events including for instance immune cell response, muscle contraction and neuronal signaling[4]. It is constituted by Orai1, the pore-forming unit of the channel located in the plasma membrane (PM) and the stromal interaction molecule 1 (STIM1), the $Ca^{2+}$ sensor residing in the membrane of the main intracellular $Ca^{2+}$ store, the endoplasmic reticulum (ER)[5,6]. After the release of $Ca^{2+}$ from the ER stores, a signaling cascade is initiated that leads to the coupling of STIM1 to Orai1 and finally to $Ca^{2+}$ influx into the cell. CRAC channel dysfunction can arise due to loss-of-function (LoF) or gain-of-function (GoF) mutations either in STIM1 or Orai1. LoF may lead to immune deficiency, which has been associated with an increased infection rate and a series of non-immunological symptoms. GoF has been linked to Stormorken-like syndrome and tubular aggregate myopathy[7].

The currently available crystal and cryo-electron microscopy (cryo-EM) structures of *Drosophila melanogaster* Orai (dOrai) consistently revealed that Orai channels form hexameric complexes, which is unique among $Ca^{2+}$ ion channels. dOrai shares 73% homology to human Orai1 (here referred to as Orai1). Each subunit contains four transmembrane domains (TM1–TM4), one intracellular (TM2–TM3) and two extracellular (TM1–TM2, TM3–TM4) loops, as well as a cytosolic N- and C- terminus[8–11]. The $Ca^{2+}$-channel pore is formed by a ring of six TM1 domains in the center of the channel complex with a diameter of 3.8–3.9 Å at the narrowest point[12], and is surrounded by TM2 and TM3 in a second ring and by TM4 helices in a third ring[9]. TM4 contains a kink in the center at P245, which has been linked to Stormorken-like syndrome (P245L)[13]. Furthermore, it is connected via the so-called nexus region to the helical C-terminus in the cytosol[9,14], forming the main coupling site for STIM1 at the periphery of the channel complex[15]. Functional studies suggest that STIM1-induced wild-type (wt) Orai1 pore opening requires a wave of interdependent TM domain motions within the entire channel complex[16]. This involves structural rearrangements along TM4 and the C-terminus, however, the extent of this conformational change is still under debate[17]. Moreover, a series of gating checkpoints in all TM domains[16] govern pore

[1]Institute of Biophysics, JKU Life Science Center, Johannes Kepler University Linz, Linz, Austria. [2]Institute for Physiology and Pathophysiology, Johannes Kepler University Linz, Linz, Austria. [3]Institute of Theoretical Physics, Johannes Kepler University Linz, Linz, Austria. [4]Department of Physics and Materials Science, University of Luxembourg, Luxembourg City, Luxembourg. [5]These authors contributed equally: Valentina Hopl, Adéla Tiffner. ✉e-mail: daniel.bonhenry@gmail.com; isabella.derler@jku.at

opening, among which some are associated with disease[18] (e.g. Orai1-A137[19], Orai1-L138[20], Orai1-V181[21], Orai1-P245[13]). Comparison of the structural resolutions of the closed state with the open state suggests a rigid-body outward movement of the individual subunits to initiate pore opening[10]. In addition, molecular dynamics (MD) simulations on the closed state of Orai1 propose a twist-to-open gating mechanism[22]. In addition to these global conformational changes, among individual TM domain movements, it has been identified that STIM1 induces a rotation of TM1 in the hydrophobic cavity region of the pore to permit $Ca^{2+}$ conduction[23].

$Ca^{2+}$ permeation through the Orai1 pore is initiated by the attraction of $Ca^{2+}$ via the $Ca^{2+}$ accumulating region (CAR) at the extracellular side[24]. Subsequently, $Ca^{2+}$ ions are forwarded to the narrow selectivity filter formed by a ring of glutamates (E106), which is followed by the hydrophobic cavity in the pore center and the basic region at the cytosolic side[25]. Hydration of the Orai1 pore is a critical prerequisite for $Ca^{2+}$ permeation[26]. Here, the hydrophobic region plays an essential role, by ensuring hydrophobic gating[23], in which the permeation of ions and water is blocked without physical occlusion of the pore. Rotation of the hydrophobic residues F99 in TM1, contributing to the hydrophobic cavity, away from the pore enhances hydration and thus $Ca^{2+}$ permeation[23]. Furthermore, the basic region in the Orai1 pore, which is decorated by positive charges, promotes hydration of the hydrophobic region via long-range effects[27]. Moreover, previous MD simulations have revealed the existence of a hydrated region at the interface between helices TM1, TM2 and TM3 in the Orai1-channel structure, referred to as the back of the pore[28]. Permeation studies using MD simulations suggest that the water molecules at the back of the TM1 helix may, in addition to the actual selectivity filter in the pore-lining helix, play a role in controlling and maintaining the selectivity of the Orai1-channel[28].

In this study, we discovered that not only the hydration within the pore, but also hydration along peripheral TM-interfaces can lead to pore opening. To this end, we investigated the molecular mechanisms underlying basic substitutions close to the middle of the TM3/TM4-interface using functional and computational studies. Herewith, we provide a deeper structural and physicochemical understanding of the Orai1 gating checkpoints for activation with a particular focus on V181 mutations close to the center of TM3 and offer insights into the physiological activation mechanism of CRAC channels.

## Results

### Charged residues at V181 in TM3 trigger pore opening

We recently discovered V181 to act as a critical and isoform-specific gating checkpoint in the Orai1-channel complex[16,29]. Remarkably, its substitution to a positively charged Lys (V181K) in Orai1 led to drastically enhanced currents compared to wt Orai1[29]. To understand the molecular underpinnings of this effect, we initially investigated the impact of a set of other canonical substitutions of different properties at this position in Orai1 (Fig. 1A–D) in the absence of STIM1. Among other charged residues incorporated at position 181, Glu (V181E) and Arg (V181R) resulted also in constitutive activity, though to a 2–3-fold reduced extent compared to V181K (Fig. 1B–E). Substitutions with Asp (V181D), Gln (V181Q) and His (V181H) left the channel in an inactive state (Fig. 1B–E). Its mutation to amino acids with small side chains (Gly (V181G), Ala (V181A), Ser (V181S)) or to the sulfur-containing Met (V181M) led to weak constitutive activity, while a Cys-substitution (V181C) left Orai1 inactive (Fig. 1C–F). Among hydrophobic substitutions, aromatic Tyr (V181Y) and Trp (V181W) substitutions led to small constitutive activity, whereas the aromatic Phe (V181F), the aliphatic Ile (V181I) and Leu (V181L) substitutions tended to keep Orai1 in the inactive state (Fig. 1D–G). Using $Ca^{2+}$ imaging microscopy, we confirmed the electrophysiologically determined significant enhancement in the constitutive activity of Orai1-V181K/E/R compared to the weaker activity of Orai1-V181A/Y and the absence of constitutive activity of Orai1-V181L (Supplementary Fig. 1A–C). Unexpectedly, Orai1-V181G showed almost as high constitutive activity as Orai1-V181K/E/R. Moreover, the difference in the magnitude of the constitutive currents of Orai1-V181K versus Orai1-V181E/R was not as pronounced in $Ca^{2+}$ imaging studies. To understand the reason

for these differences, we determined the change in local $Ca^{2+}$ concentration of these Orai1-mutants by means of a C-terminally attached $Ca^{2+}$-indicator GCamp6f and a stepwise increase in extracellular $Ca^{2+}$ concentration. In this way, we were able to determine a stepwise increase in cellular $Ca^{2+}$ levels for Orai1-V181K/R/E/A/G/Y-GCamp6f expressing cells, the magnitude and relative changes of which were more consistent with detected $Ca^{2+}$ currents of the respective mutants (K < R < E < A < G < Y) (Supplementary Fig. 1D–F). Taken together, Orai1-V181K showed the strongest constitutive activity in the absence of STIM1.

All tested V181X mutants revealed either store-operated or constitutive Orai1-channel activity in the presence of STIM1, except for V181F (Fig. 1E–G and Supplementary Fig. 1G–I). Interestingly, Orai1-V181K/E/R currents did not drastically enhance after store-depletion in the presence of STIM1, while Orai1-V181Q/D exhibited weak store-operated activation. The current/voltage (I/V) relationship of constitutively active Orai1-mutants, as exemplarily shown for Orai1-V181K, Orai1-V181A and Orai1-V181W, exhibited the typical inwardly rectifying characteristic already without STIM1 co-expressed as known for CRAC channel currents[30] (Fig. 1H–J). The reversal potentials ($V_{rev}$) of the Orai1-mutants were in the range of +40 to +50 mV. In the presence of STIM1, the I/V-relationships exhibited a rightward shift of the $V_{rev}$ in the range of +50 to +60 mV, comparable with authentic CRAC channel currents[30].

Complementary to our electrophysiological studies, we investigated the Orai1-V181 mutants for their ability to induce NFAT (nuclear factor of activated T-cells) translocation in the absence of STIM1 (Fig. 1K, L and Supplementary Fig. 2A–D). NFAT-translocation occurs after elevations of intracellular $Ca^{2+}$ concentrations triggered by $Ca^{2+}$-channels and calmodulin to activate the phosphatase calcineurin. The latter dephosphorylates NFAT leading to the translocation of NFAT from the cytosol into the nucleus to induce gene regulation[31–33]. In accord with observed current levels, Orai1-V181K led to robust NFAT-translocation to almost 100%. Similarly, other constitutive mutants with charged substitutions (Orai1-V181E/R) exhibited NFAT-translocation in the range of 90–100% (Supplementary Fig. 2A–D). Weak constitutively active Orai1-mutants (Orai1-V181A/S/W) exhibited NFAT-translocation in the range of 20–30% (Supplementary Fig. 2B–D). As expected, the inactive Orai1-V181L/I/F/Q-expressing cells showed almost no detectable NFAT-translocation to the nucleus similar to wt Orai1 in the absence of STIM1 (Supplementary Fig. 2A, C, D). Unexpectedly, Orai1-V181G and Orai1-V181Y exhibited in 70% or 60% of the tested cells, respectively, NFAT-translocation (Supplementary Fig. 2B–D) despite weak constitutive current densities comparable to Orai1-V181A/S/W (Fig. 1C, D). To understand whether prolonged exposure of the cells to $Ca^{2+}$-containing media could be the reason for the drastically increased NFAT-translocation in the case of some weaker active Orai1 GoF mutants (Orai1-V181E/R/G/Y), we performed time-dependent NFAT-translocation experiments in which cells were incubated in $Ca^{2+}$-free buffer prior to the experiment. Solution exchange of a 0 mM $Ca^{2+}$- with a 2 mM $Ca^{2+}$-containing buffer was performed at the beginning of the experiment (Supplementary Fig. 2E–G). Observation of NFAT-translocation over 6 h showed that Orai1-V181K/E/R started as expected at a higher level, which further increased to a maximum within the first 2 h. Orai1-V181A/G/Y caused slightly enhanced NFAT-translocation, while Orai1-V181L/W remained at low levels comparable to wt. Subsequent ionomycin (10 μM) treatment drastically enhanced NFAT-translocation to 100% (Supplementary Fig. 2E–G). Overall, NFAT-translocation levels correlated with the $Ca^{2+}$ currents (Fig. 1B–D) and $Ca^{2+}$ entry (Supplementary Fig. 1A–F) of the respective Orai1-mutants, except for the comparable levels of Orai1-V181K/E/R. The latter could still be due to the excessive $Ca^{2+}$ concentration in Orai1-V181E/R expressing cells.

In the presence of STIM1 and before store-depletion, NFAT-translocation levels were comparable to those in the absence of STIM1, but increased to maximum levels upon store-depletion comparable to wt conditions (Supplementary Fig. 2H–J). Hence, all mutants can be activated upon co-expression with STIM1 and store-depletion to levels observed under wt conditions (Supplementary Fig. 2H–J).

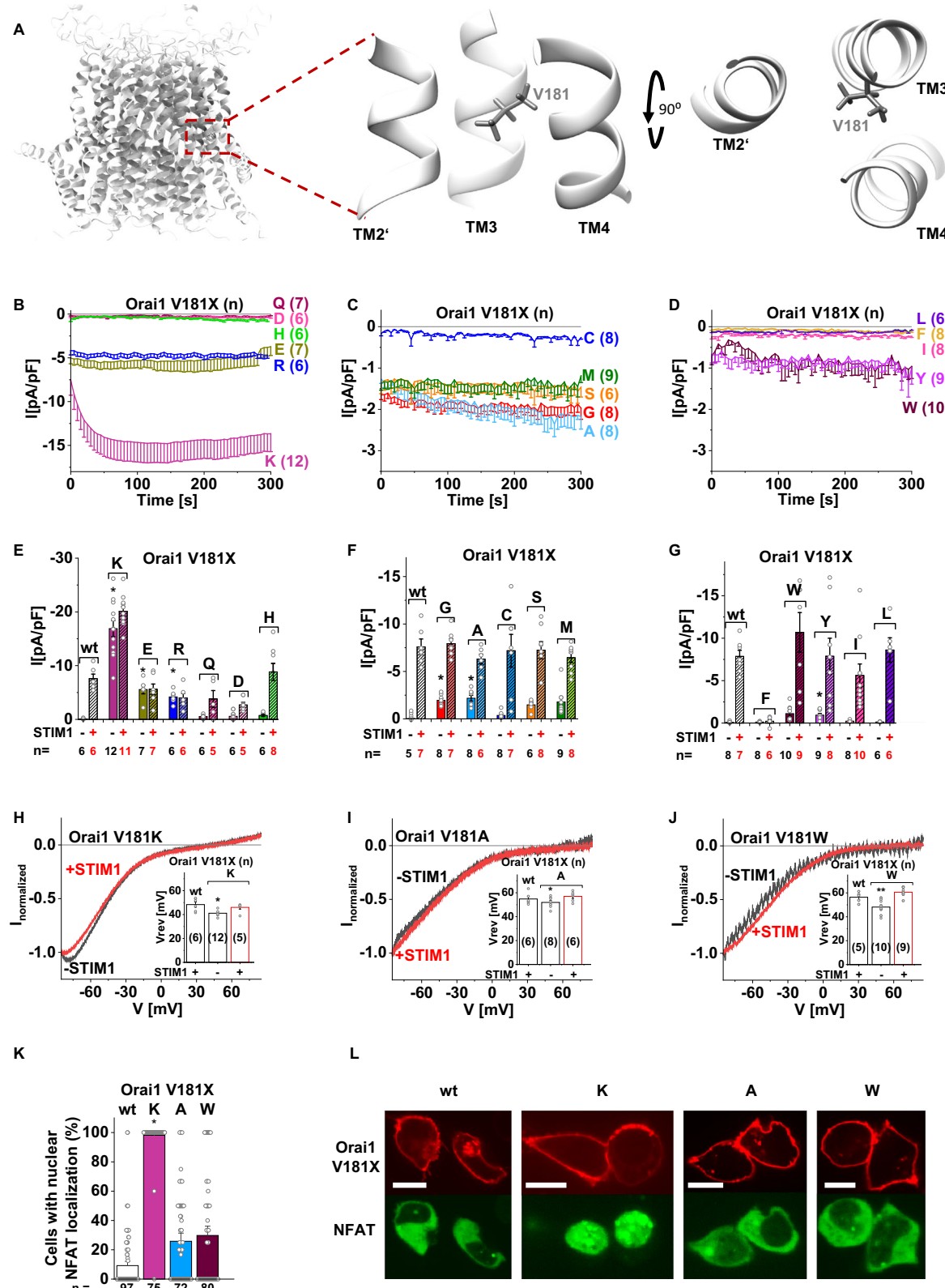

Using confocal FRET microscopy, we investigated the most critical Orai1-V181X mutants containing amino acid substitutions with charged (K, E, R, D), small (A, G) and hydrophobic (I, Y, L, W) side chains for coupling to STIM1. Most of the tested mutants showed a clear increase in FRET upon store-depletion to comparable levels like wt (Supplementary Fig. 3A–D). An exception is Orai1-V181D as it shows only a weak increase

in FRET (Supplementary Fig. 3A), which is in accord with its decreased activity in the presence of STIM1 (Fig. 1E and Supplementary Fig. 1G). In contrast to wt Orai1 and Orai1-V181K, Orai1-V181E/R exhibited slightly slower store-dependent increase in FRET (Supplementary Fig. 3A). Additionally, we examined the homomerization FRET of C-terminally CFP-/ YFP-labeled Orai1-V181K/E/R. Our results revealed that the degree of

**Fig. 1 | Exchanging Orai1-V181 by residues with charged side chains triggers robust constitutive activity. A** Schematics showing a side view of the entire Orai1-channel complex as well as side and top views of the TM3/TM4-interface along with TM2′ of the next subunit highlighting the tested residue V181. Time courses of $Ca^{2+}$ current densities after whole-cell break-in of Orai1-V181X (X = K/E/R/Q/D/H) (**B**), (X = G/A/C/S/M) (**C**), (X = F/W/Y/I/L) (**D**) in the absence of STIM1. **E–G** Block diagrams of maximal whole-cell current densities of mutants recorded in (**B–D**) compared to Orai1 (wt) in the absence (−) and presence (+) of STIM1. Normalized current/voltage (I/V) relationships of Orai1-V181K (**H**), Orai1-V181A (**I**), and Orai1-V181W (**J**) in the absence (−) and presence (+) of STIM1 were taken at maximum current levels. Insets represent the Vrev of wt STIM1 + Orai1 currents versus the respective constitutive mutant currents. Single values are indicated as gray circles. **K** The average number of HEK293 cells that exhibit NFAT localization to the nucleus determined upon co-expression (CFP-NFAT) with YFP-Orai1 (wt), or corresponding mutants shown in (**H–J**) after 24 h in 2 mM $Ca^{2+}$-containing media. For the analysis, 72–97 images of cells containing in total 72–328 cells were used. **L** Representative images of HEK293 cells co-expressing YFP-Orai1 (wt) or corresponding mutants shown in (**K**), respectively, with CFP-NFAT in the presence of 2 mM $Ca^{2+}$ (Scale bar, 10 μm). For the presented bar diagrams, ANOVA (one-way/Kruskal–Wallis ANOVA) or Welch-ANOVA was employed for statistical analyses with differences considered statistically significant at $p < 0.05$ (see Supplementary Data 1). Statistical differences of wt Orai1 compared to all Orai1-V181X mutants in the absence of STIM1 is indicated with asterisks.

homomerization between Orai1-V181X-CFP and Orai1-V181X-YFP (X = E, K, R) significantly decreased by 25–50% compared to wt Orai1-CFP – wt Orai1-YFP and Orai1-V181A-CFP – Orai1-V181A-YFP (Supplementary Fig. 4A). These changes in STIM1-Orai1 or Orai1-Orai1 FRET upon V181 substitutions match the altered activation properties of Orai1 mutants in the presence of STIM1.

We have previously identified that a decrease in the hydrophobicity of TM3 correlates with an increase in the constitutive activity of Orai1-mutants[29]. Accordingly, we performed bioinformatic analysis of hydrophobicity profiles of Orai1-V181K/R/E/F/A, using a prediction program (see section "Methods") based on the Roseman hydrophobicity scale[34]. The hydrophobicity profile of Orai1 and corresponding mutants indicated the formation of four TM domains in correlation with recent studies[29] (Supplementary Fig. 4B top). As expected, overall hydrophobicity along TM3 decreased for constitutively active mutants, while the inactive Orai1-V181F exhibited enhancement (Supplementary Fig. 4B bottom).

Summarizing, substitutions of V181 within the TM3/TM4-interface by the residues K, R and E with charged side chains trigger strong constitutive activity, which does not or only marginally affect STIM1-coupling but likely impacts the conformation of peripheral channel segments around the C-termini.

### Charged residue at A254 in TM4 in close proximity to V181 in TM3 triggers pore opening

Based on the unique effect of Orai1-V181K, we next screened through other positions along the TM3/TM4-interface by polar amino acid substitutions. Indeed, we discovered that Orai1-A254K/E/Q in TM4 directly opposite to V181 (Fig. 2A) triggered constitutive currents and NFAT-translocation already in the absence of STIM1 (Fig. 2B–F). In the presence of STIM1, constitutive currents enhanced slightly, but not significantly for Orai1-A254K/Q and remained comparable for Orai1-A254E (Fig. 2C and Supplementary Fig. 4C). As exemplarily shown for Orai1-A254K, constitutive currents exhibited CRAC channel-typical inward rectification with the $V_{rev}$ in the range of +50 mV, both, in the absence as well as the presence of STIM1 (Fig. 2D). Using FRET microscopy, we detected only weak increase in STIM1-binding to the constitutive Orai1-mutants containing a charged substitution at A254, as exemplarily shown for Orai1-A254K/E, upon store-depletion, which was significantly reduced compared to wt conditions (Supplementary Fig. 3E, F). This is consistent with the weak or lack of STIM1-mediated current enhancements. In analogy to TM3 mutants, the hydrophobicity profile along TM4 decreased for the constitutively active Orai1-A254E/K/Q mutants (Supplementary Fig. 4D). Remarkably, Orai1-A254E showed only weak PM expression (Fig. 2F) despite strong constitutive activity. This suggests that the charged side chain at position 254 in Orai1 has a substantial effect, enabling even a small amount of the protein in the PM to induce strong activation. However, this finding also indicates that a charged side chain at some positions in the TM region can also hinder PM expression.

To investigate the impact of charged substitutions along Orai1 TM-interfaces on Orai1 function, we screened through several other positions. Substitutions to charged amino acids (especially Lys) at A177 (Orai1-A177K) and F253 (Orai1-F253K) close to V181 and A254 along TM3 and TM4, respectively, showed solely store-operated activation in the presence of STIM1 (Supplementary Fig. 4E, F). Other positions in TM3 mutated to charged residues exhibited no or store-operated activation (Supplementary Fig. 4E). Evaluation of the mutants PM expression revealed for most of them as well as other substitutions also in TM2 (L130R, F136R) at least partially impaired PM localization in contrast to Orai1-V181K or Orai1 as exemplarily shown for Orai1-L130R, Orai1-F136R, Orai1-L185K-F250K, Orai1-F187K, Orai1-A189K, Orai1-L193K and Orai1-C195K (Supplementary Fig. 4G). This highlights the uniqueness of Orai1-V181K and Orai1-A254K in maintaining PM expression, at least partially, and triggering GoF.

Summarizing, only substitutions of V181 or A254 within the TM3/TM4-interface by residues with charged side chains preserve PM expression and trigger strong constitutive activity.

### Charged substitutions at V181 alter the structural stability of Orai1-mutants

To obtain a better understanding of the robust constitutive activity of Orai1-V181K compared to the different effects of other V181 substitutions (Orai1-V181E/R/A/F), we performed MD simulations using a homology model of Orai1 (Supplementary Fig. 5A, B)[24] based on the crystal structure of dOrai (Protein Data Bank ID: 4HKR)[9] representing the inactive state. Additionally, we investigated protonated Orai1-V181E (termed Orai1-V181E⁰) to focus entirely on the impact of locally modified electrostatic interactions, while the effects on the volume occupied by the side chain remain negligible. To prove the quality of the MD simulations we show two movies of Orai1 side and top view (Supplementary Movies 1 and 2).

Initially, we evaluated separately the root mean square deviation (RMSD) of all four TM domains of the Orai1-mutants (Orai1-V181A/F/E⁰/K/E/R) in comparison to wt Orai1 (Supplementary Fig. 5C–I). Wt Orai1 reached equilibrium at about 250–300 ns with a Cα-RMSD in the range of 2.7 Å for TM1 and 1.8–2.2 Å for TM2–TM4, which is in line with our previous work[35].

In contrast, the mutants did not show the same level of structural stability, with RMSD values still increasing at 250 ns and reaching equilibrium only around 300–400 ns, except Orai1-V181E. For the latter mutant, an extension of the simulation time to 550 ns was required to reach equilibrium. Orai1-V181A exhibited a similar time to reach equilibrium and Cα-RMSD comparable to wt, with the former shifting to 1.8–2.2 Å for TM1–TM4 (Supplementary Fig. 5D). Orai1-V181F also presented a slow but steady increase in RMSD values around 250 ns for all the TM domains (Supplementary Fig. 5E). The electrostatically neutral Orai1-V181E⁰ modification exhibited a comparable behavior like wt Orai1 (Supplementary Fig. 5F). The mutation V181K had the most significant effect on the conformational flexibility of the protein with overall RMSD values higher than that of any other mutant simulated (Supplementary Fig. 5G). Mutation to a glutamic acid (V181E) also affected the channel but to a lower extent compared to the lysine mutant (V181K) and was unable to reach equilibrium even after 400 ns, but after 550 ns (Supplementary Fig. 5H). Yet, the perturbating effect of an electrostatic charge at position 181 on TM2, TM3 and TM4 can be seen, particularly when compared to the protonated glutamic acid. The RMSD values of all TMs for V181E increased by almost 1 Å around the 350–400 ns mark compared to V181E⁰. Surprisingly, the

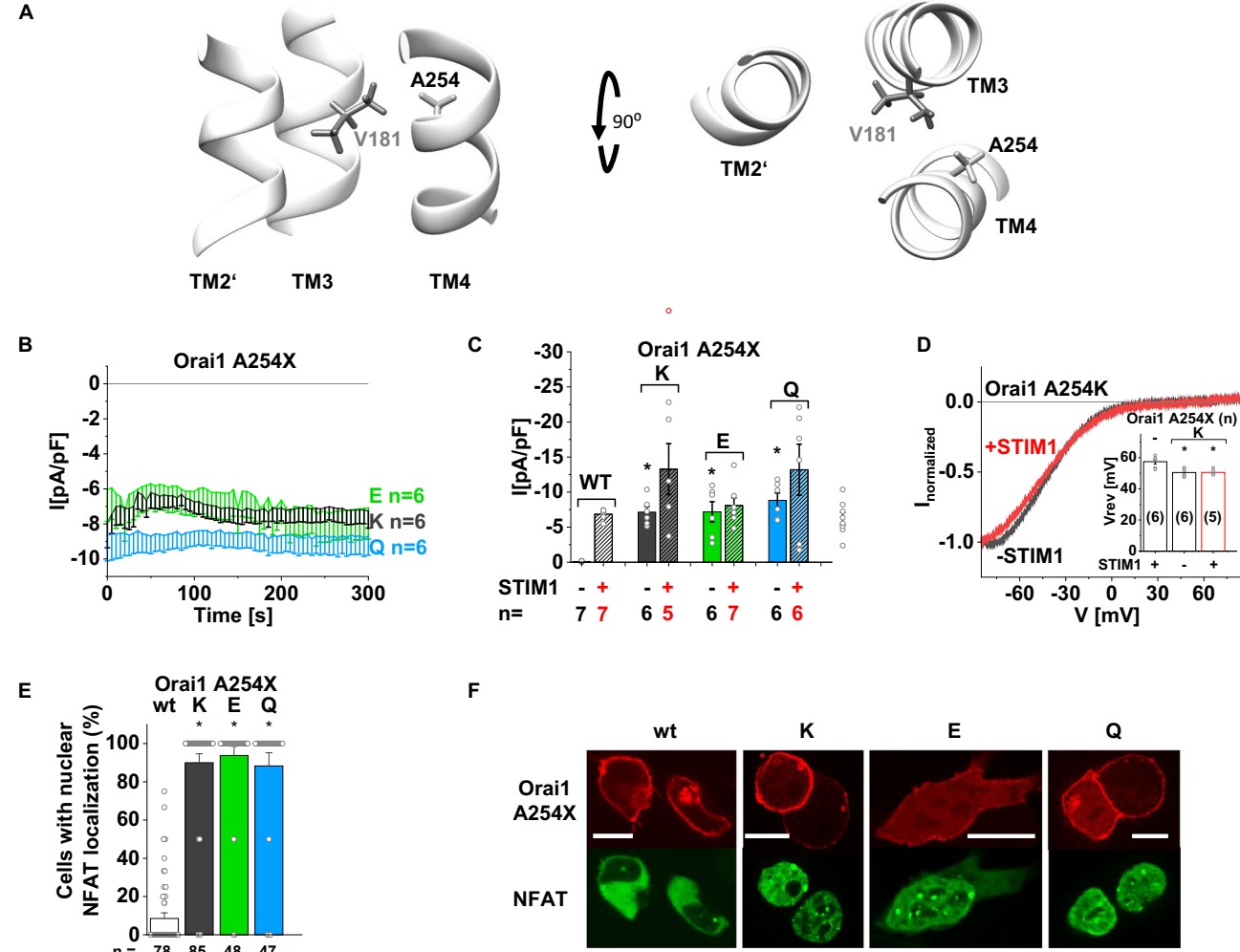

**Fig. 2 | Substitution of Orai1-A254 by charged residues triggers constitutive activity. A** Schematics showing side and top views of the TM3/TM4-interface along with TM2′ of the next subunit highlighting the tested residue A254 (white) opposed to V181 (light gray). **B** Time courses of Ca²⁺ current densities after whole-cell break-in of Orai1-A254X (X = K/E/Q) in the absence of STIM1. **C** Block diagram of maximal whole-cell current densities of mutants recorded in (**B**) compared to Orai1 (wt) in the absence (−) and presence (+) of STIM1. **D** Normalized current/voltage (I/V) relationships of Orai1-A254K in the absence (−) and presence (+) of STIM1 were taken at maximum current levels. Inlet represents reversal potential (Vrev) of wt STIM1 + Orai1 currents versus respective constitutive mutant currents. Single values are indicated as gray circles. **E** The average number of HEK293 cells that

exhibit nuclear NFAT localization determined upon co-expression (CFP-NFAT) with YFP-Orai1 (wt) or YFP-Orai1-A254X (X = K/E/Q) mutants in the absence of STIM1 after 24 h in 2 mM Ca²⁺ containing media. For the analysis, 47–85 images of cells containing in total 72–245 cells were used. **F** Representative images of HEK293 cells co-expressing corresponding mutants with CFP-NFAT in the presence of 2 mM Ca²⁺ (Scale bar, 10 μm). For the presented bar diagrams, Welch-ANOVA and Kruskal–Wallis ANOVA were employed for statistical analyses with differences considered statistically significant at $p < 0.05$ (see Supplementary Data 1). Statistical differences of wt Orai1 compared to all Orai1-A254X mutants in the absence of STIM1 is indicated with asterisks.

introduction of an arginine at position 181 had only a very limited perturbating effect, especially considering the effect of the lysine (Supplementary Fig. 5I), which is a quite close analog to an arginine compared to the other mutations. Among the mutants showing higher constitutive activity than Orai1-V181A, Cα-RMSD augmented to around 2.7–3.7 Å. However, they do not present higher variation denoting a faint effect on the structure.

To find out whether the experimentally determined functional status of the Orai1-mutants was reflected in the pore radius, we calculated the latter (Fig. 3A). Orai1-V181F exhibited a slightly enhanced pore diameter around the selectivity filter, but otherwise a comparable pore radius profile along the pore axis from the extra- to the intracellular side like wt, in line with LoF or maintenance of the resting state. In contrast, the constitutively active Orai1-V181X (X = K, R, E, A) mutants exhibited an increased pore radius around the selectivity filter and even more pronounced dilation toward the cytoplasmic side. The extent of the increase in pore radius correlated with the levels of constitutive currents (Fig. 3A). The enhanced pore radius around

the selectivity filter increases the likelihood that ions can bind and enter the pore of the mutant and is in accord with the slightly, but significantly decreased $V_{rev}$ (Fig. 1H–J). Further dilation toward the basic region is crucial for higher hydration, a reduction of steric hindrance as well as a mitigation of electrostatic exclusion between basic residues and cationic ions. The expansion of the basic region seems to represent a key step for the activation of the constitutively active mutants in accord with previous findings[8,10,11,27].

Summarizing, the introduction of different mutations, albeit at the same site, appears to affect the structure of Orai1 and its overall conformational dynamics in different ways.

## Charged side chains at position 181 in Orai1 enhance pore hydration

The hydrophobic gate and the basic region in the Orai1 pore are key determinants contributing to the control of its opening[26]. The latter tunes the level of pore hydration at the hydrophobic gate to allow the Orai1-

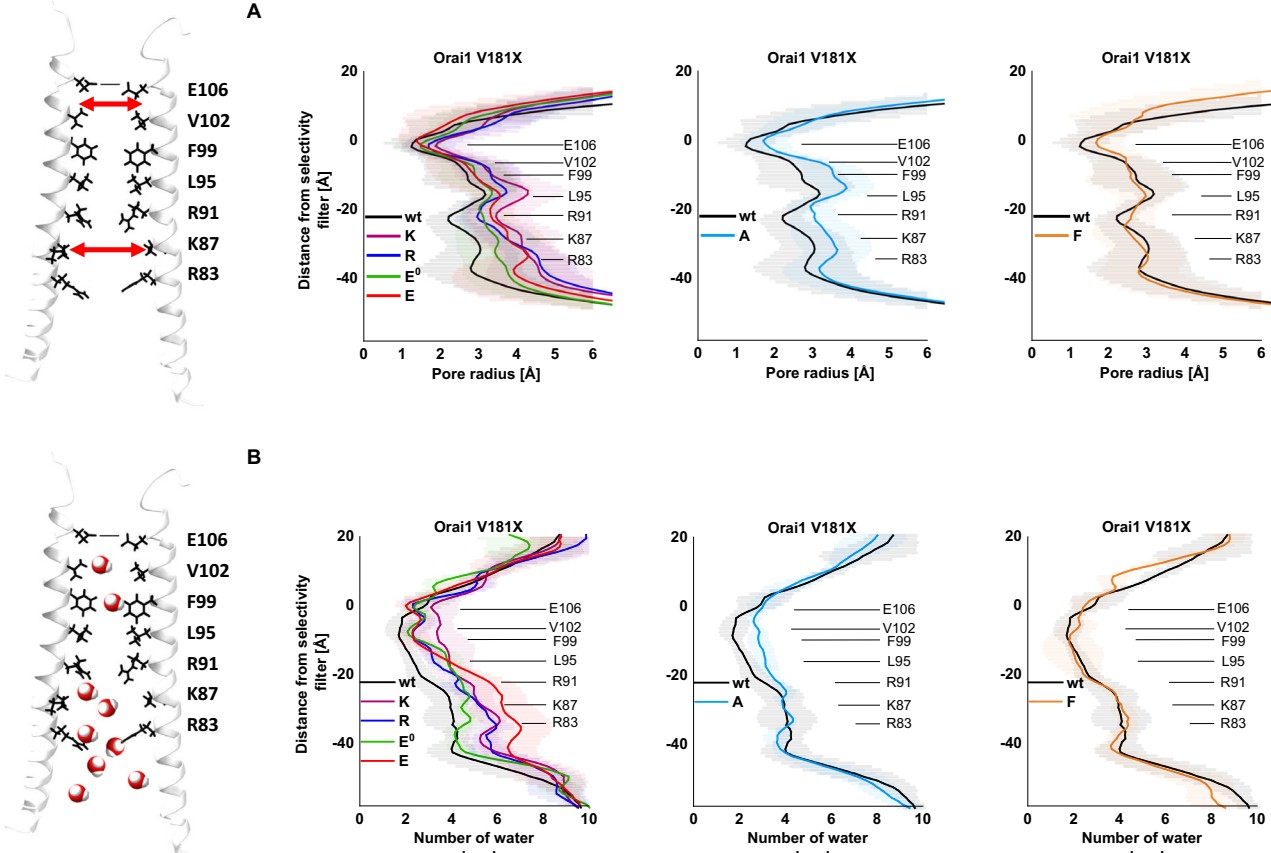

**Fig. 3 | Charged residues at position 181 in Orai1 trigger strong pore dilation and hydration.** The scheme depicts pore-lining residues of the Orai1-channel. The particular positions are indicated on the right. Red arrows (top) indicate pore diameter, as the pore radius was calculated in (**A**). Water molecules along the pore (bottom) indicate that in (**B**) the level of pore hydration was determined. **A** The cavity size of the pore as calculated by HOLE is given. The plots use the following color code, V181K (purple), V181R (blue), V181E (red), V181E$^0$ (teal), V181A (cyan), V181F (orange) and Orai1 (wt; black). The shaded areas correspond to the statistical error over the three replicas. Simulation data from 350 to 400 ns was used. **B** Distribution of water oxygen along the *z*-axis within 10 Å from the center of the pore. The plots are using the same color code as in (**A**). Simulation data from last 50 ns was used. The shaded areas correspond to the standard deviation over the three replicas. For (**A**) and (**B**), the location of the pore-lining residues R83, K87, R91, L95, F99, V102 and E106 are indicated.

channel to close or open[27]. Since pore hydration is a crucial indicator for the extent of ion channel conductance, we investigated the difference in wetting along the pore of the mutants compared to wt Orai1 (Fig. 3B). Overall, the hydration profiles of the mutants follow the same trend as the pore radius profiles. For wt Orai1, hydration is lowest around the selectivity filter (E106), while it enhances toward the extracellular as well as intracellular side (Fig. 3B). The mutants Orai1-V181F exhibited a hydration profile comparable to that of wt Orai1. In contrast, Orai1-V181E$^0$ and the constitutively active Orai1-V181X (X = A, E, R, K) mutants showed enhanced hydration along the pore profile. Orai1-V181E$^0$ and Orai1-V181A exhibited a higher hydration only around the hydrophobic region, but was otherwise comparable to wt Orai1. Orai1-V181R and Orai1-V181E exhibited higher hydration around the basic region, while Orai1-V181K exhibited increased hydration along the entire pore region compared to wt Orai1. Enhanced hydration results in a reduced free-energy barrier[36] for ion passage and its extent is in qualitative agreement with the current density levels measured for the respective mutants. To sum up, charged residues close to the center of TM3 enhance hydration along the pore in line with enhanced permeability for $Ca^{2+}$ ions.

### Charged substitutions at position 181 in Orai1 enhance hydration in the TM2/TM3/TM4-interface

Previously, MD simulations of wt Orai1 have shown a hydrated region at the interface of TM1 and the TM2/TM3 ring[28]. Our investigation revealed that hydration around position 181 was drastically enhanced for Orai1-V181K

(Fig. 4A–C), from about 3 to ~18 water molecules per channel occupying regions in the range of 6 Å surrounding the Cα of 181 (Fig. 4D). Orai1-V181E$^0$ and Orai1-V181R exhibited similarly enhanced hydration in the range of 13–18 water molecules on average like Orai1-V181K. Interestingly, Orai1-V181E exhibited even higher back pore hydration. This more pronounced hydration for V181E could possibly be explained by the relatively small size of the glutamic acid side chain compared to arginine and lysine. The latter two possibly occupy the positions that would be occupied by water molecules in the case of glutamic acid. Inactive (Orai1-V181F) and weakly constitutively active (Orai1-V181A) mutants exhibited a comparable number of water molecules like wt Orai1 (Fig. 4D and Supplementary Fig. 6A–F). It seems that primarily basic or acidic residues are able to recruit more water molecules to the peripheral TM-interface as well as to the entire channel complex, as revealed by the analysis of the distribution of water oxygen along the *z*-axis with a radius of 30 Å (Fig. 4C).

A closer look at the surrounding of V181 and K181 indicated that especially the residues C143 in TM2, A177 in TM3 and F253 in TM4 are directed to the water crevice around position 181 (Fig. 4E). In particular, for the mutants Orai1-V181K and Orai1-V181E, the area formed by these three positions tended to increase (Fig. 4F, G). Interestingly, this area increased also in Orai1 V181F, while leaving the channel inactive, potentially due to its water-shielding effect, thus, maintaining a dewetted TM3/TM4-interface. Overall, the constitutive activity of Orai1-mutants containing a charged side chain at position 181 in TM3 is accompanied by enhanced hydration around this position.

**Fig. 4 | Charged residues at position 181 in Orai1 trigger enhanced hydration along the TM3/TM4-interface.** Left: side view of representative snapshots for Orai1 (**A**) and Orai1-V181K (**B**) depicted as gray ribbon using a transparent representation with residue 181 shown as sticks in gray and blue, respectively, taken from an equilibrated trajectory part at $t = 400$ ns. The positions occupied by oxygen atoms were selected around 6 Å from the Cα from residue 181 and 6 Å above and below from the $z$ position of residue 181. Each gray bead represents an oxygen atom from the water molecules from a superposition of 25 snapshots covering 50 ns between 350 and 400 ns simulations. Middle: bottom view of representative snapshots from the channel. Right: representative snapshots of a subunit and the water molecules around 6 Å from residue 181. Water molecules around the residues are depicted using red balls for the oxygen atoms and white balls for the hydrogen atoms. **C** Hydration profile for the back pores. Water molecules in a cylinder of radius of 30 Å centered on the main pore of the protein (defined as the Cα from residues 80 to 110) were selected for calculation. The last 50 ns are used. The distributions use the following color code, V181K (purple), V181R (blue), V181E (red), V181E⁰ (teal), V181A (cyan), V181F (orange) and Orai1 (wt; black). Shaded areas correspond to the standard deviation of the mean over three replicas. The gray area indicates the standard deviation assigned to the position of the V181 side chain. The location of the pore-lining residues R83, K87, R91, L95, F99, V102 and E106 are indicated. **D** Distributions of the number of water molecules around 6 Å from the Cα of residue 181 for the wild-type and its mutants for all the replicas. The last 50 ns were used. The distributions use the following color code, V181K (purple), V181R (blue), V181E (red), V181E⁰ (teal), V181A (cyan), V181F (orange) and wt (gray). **E** Bottom view (left) and side membrane view (right) of a representative snapshot from a subunit from V181K. The TM segments are represented as gray ghostly ribbons with residues C143, A177, K181 and F253 shown as green, blue, purple and orange sticks. Snapshots have been taken from an equilibrated trajectory part at $t = 400$ ns. **F** Bottom view of the wt Orai1 (homology model). TM1 (residues 80–110), TM2 (residues 120–150) and TM3 (residues 170–195) are depicted as red, green and blue transparent ribbons respectively. TM4 (residues 235–262) is depicted as a solid gray ribbon. Cα from residues 143, 177 and 253 are shown as yellow beads. The triangular area formed by these three atoms is depicted in orange. **G** Bar plot for the areas formed by the Cα from residues 143, 177 and 253 for the wild-type and its mutants during the last 50 ns. Mean values over three different replicas and standard deviation of the mean are indicated by error bars. Areas for the wt Orai1 (gray) compared to Orai1-V181X (X = A, F, K, R, E⁰, E) are colored in cyan, orange, blue, purple, teal and red, respectively.

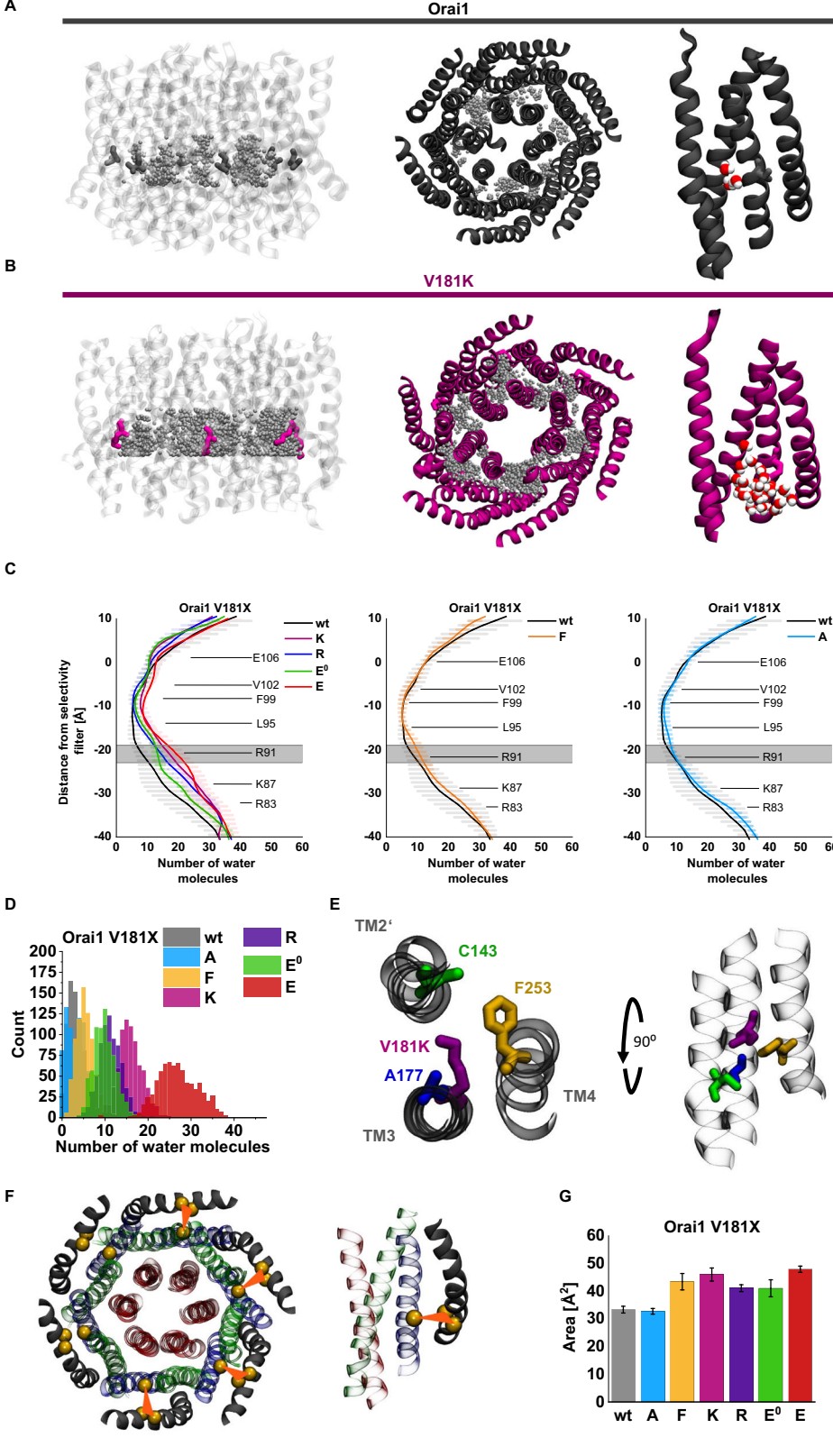

## V181 substitutions have structural effects on the peripheral TM-interface and the channel complex

To gain a better insight into the effects of the respective mutations on the structure, we looked at the reorientation of the different residues by calculating the angle between the vectors projected in the plane of the membrane formed by the Cα with the Cγ of the residue at position 181 and by the Cα from residue 181 and the center of the pore (Fig. 5A, B). For all mutants containing charged substitutions, the distributions are shifted toward lower values, corresponding to a reorientation of the side chains toward the pore. The distributions for V181K, V181R and V181E are comparable. It appeared that the broadest distribution compared to the wt Orai1 occurs in the Orai1-V181E$^0$ mutant with a net reorientation toward the pore.

Based on this observation, we also calculated the distances between the geometrical center of the charged moieties and the center of the pore (Fig. 5A, C and Supplementary Fig. 7A–E). As a result of these calculations, it appeared that the Lys (V181K) side chain is closer to the pore than Arg (V181R), deprotonated Glu (V181E), protonated Glu (V181E$^0$) and Val (wt Orai1). The possibility of the longer and more flexible lysine to project more easily its charge toward the pore compared to a shorter glutamic acid, enables it to interact with and attract water molecules in this critical region thus enhancing the local hydration. Hence, we postulate that the presence of an electrostatic charge at this position is able to recruit more water molecules thus increasing the hydration within the pore. The case of the Orai1-V181R mutant might appear surprising considering the similarities between a lysine and an arginine side chain, same charge polarity, identical length, and flexible configuration. In simulations, to account for this observation, it appeared that the arginine side chain, rather than following the same mechanism as the homologous electrostatically charged residue is partially engaging in cation–π interactions with an aromatic residue located in close proximity, F257 (Supplementary Fig. 7E, F).

The mutant V181F presents a bimodal distribution of the rotation angle formed by the Cα with the Cγ (Fig. 5B). During the simulations, F181, which was initially wedged between TM2 and TM3, is now reoriented toward the membrane (as denoted by the two dihedral modes shown in Supplementary Fig. 7G). The bimodal distribution of the orientation angle is due to the two different configurations adopted by the phenylalanine with the higher values corresponding to residues protruding toward the membrane (Supplementary Fig. 7G).

Based on these simulations, we hypothesized that the GoF induced by a lysine at position 181 is related to the conformational flexibility and unique physical properties of its amine moieties, allowing it to directly impact the hydration in the pore.

## Site-directed mutagenesis in the surrounding of V181K modulates constitutive activity

Since our observations showed that V181K is structurally surrounded by the residues C143 (TM2′), A177 (TM3), F253 and F257 (TM4), we hypothesized that the replacement of these residues with more hydrophobic amino acids could reduce constitutive activity due to a potentially suppressed water accessibility of the contact interface. Interestingly, replacement of C143 or F253 in Orai1-V181K by small hydrophobic Ala (Orai1-C143A-V181K, Orai1-V181K-F253A) significantly reduced or abolished the constitutive activity, whereas larger hydrophobic amino acids left the constitutive activity of Orai1-V181K unaffected (Orai1 C143F/L V181K) or even led to an increase (Orai1-C143W-V181K, Orai1-V181K-F253W) (Fig. 6A–E). In contrast, large hydrophobic amino acids at either A177, F257 or both decreased or abolished the constitutive activity (Orai1-A177L/F/W-V181K, Orai1-V181R/K-F257W, Orai1-A177W-V181K-F257W), while small hydrophobic amino acids left the constitutive activity unaffected (Orai1-V181K-F257A, Orai1-V181R-F257A/V, Fig. 7A–E). Contrary to these results, overall hydrophobicity around V181K should always be increased with mutation to residues with increasing hydrophobicity. This is suggested by hydrophobicity plots showing a correlation between increased overall hydrophobicity along the respective TM domains and increasing

hydrophobicity of the amino acids exchanged therein (Supplementary Fig. 8C–F and Supplementary Fig. 9C–F).

The opposite effects of C143- and F253- compared to A177- and F257-substitutions might arise due to their location relative to V181K. While C143 and F253 are in the same membrane plane opposite V181K, A177 and F257 are located along the TM3/TM4-interface one helical turn closer to the cytosol than V181K (Figs. 6A and 7A). Hence, a drastic reduction in Orai1-V181K currents due to small amino acid substitutions in the same plane might arise due to the narrowing of the area surrounding V181K, while bulkier side chains may widen this area. This might modulate the accessibility of V181K to water molecules. Hydrophobic substitutions in the plane one helical turn closer to the cytosol may limit the accessibility of water molecules to V181K with increasing side chain size and hydrophobicity.

Co-expression of STIM1 mostly further enhanced currents in a store-dependent manner or left the constitutive activity unaffected (Supplementary Figs. 8A, B, D, E and 9A, B, D, E). A similar trend in current density levels was observed depending on the side chains like in the absence of STIM1. Interestingly, Orai1-C143A-V181K, Orai1-A177W-V181K and Orai1-A177W-V181K-F257W remained inactive in the presence of STIM1 (Supplementary Figs. 8E and 9B, E). Corresponding single point mutations at C143, A177, F253 and F257 did not abolish STIM1-mediated Orai1 activation (Supplementary Figs. 8G, H and 9G, H). Hence, observed LoF of above described double or triple mutants either in the absence and/or in the presence of STIM1 seems to be the result of an allosteric effect of the individual substitutions, which in some cases also cannot be overcome by STIM1.

To ensure that the observed effects of mutations around V181 are specific to V181K and not general to other constitutively active mutants, we investigated their effects in the known GoF-mutants Orai1-V102A[37], Orai1-H134A[35] and Orai1-F136S[38]. While F253A and C143A reduced or abolished Orai1-V181K currents, respectively (Fig. 6B–E), they did not or only slightly, but not significantly, reduce the constitutive activity of Orai1-V102A, -H134A and -F136S (Supplementary Fig. 8I–N). In contrast, C143W and F253W, which increased the constitutive activity of Orai1-V181K (Fig. 6B–E), decreased the currents of Orai1-V102A and Orai1-H134A (Supplementary Fig. 8I–N). Furthermore, while A177W, F257W and A177W F257W abolished the constitutive activity of Orai1-V181K (Fig. 7B–E), the constitutive activity of Orai1-V102A remained unaffected (Supplementary Fig. 9I, L) and the constitutive activity of Orai1-H134A was only reduced (Supplementary Fig. 9J, M), but not abolished. The constitutive activity of Orai1-F136S remained unaffected by A177W and was only abolished by F257W (Supplementary Fig. 9K, N). F257A left constitutive activity of Orai1-V181K (Fig. 7D, E) as well as of Orai1-V102A and Orai1-H134A unaffected (Supplementary Fig. 9I, J, L, M). Overall, the effects of the C143-, A177-, F253- and F257-substitutions cannot be generalized, but rather appear to be specific to Orai1-V181K.

In addition to LoF mutants with a specific effect on Orai1-V181K, we published in Tiffner et al.[16] a series of more generalized LoF mutations, interfering with the function of several GoF mutants as well as wt Orai1. Similarly, we found that an aromatic substitutions at T180 in TM3, a position pointing toward TM2 of the next subunit (TM2′) (Supplementary Fig. 10A), has more generalized effects. We discovered that T180F/W in TM3 blocks both Orai1 and Orai1-V181K activation (Supplementary Fig. 10B–D). These findings indicate that this substitution disrupts a critical gating pathway to the pore, as has already been shown for a number of other positions[16].

Recapitulating, large hydrophobic substitutions in the plane of V181K preserve or even enhance the constitutive activity, while large hydrophobic substitutions one helical turn closer to the cytosol decrease or abolish constitutive activity.

## Discussion

Hydrophobic collapse is responsible for the rapid, but not necessarily complete de-wetting of hydrophobic residues in protein structures[39,40]. This in conjunction with optimal hydration of solvent-exposed charged or polar

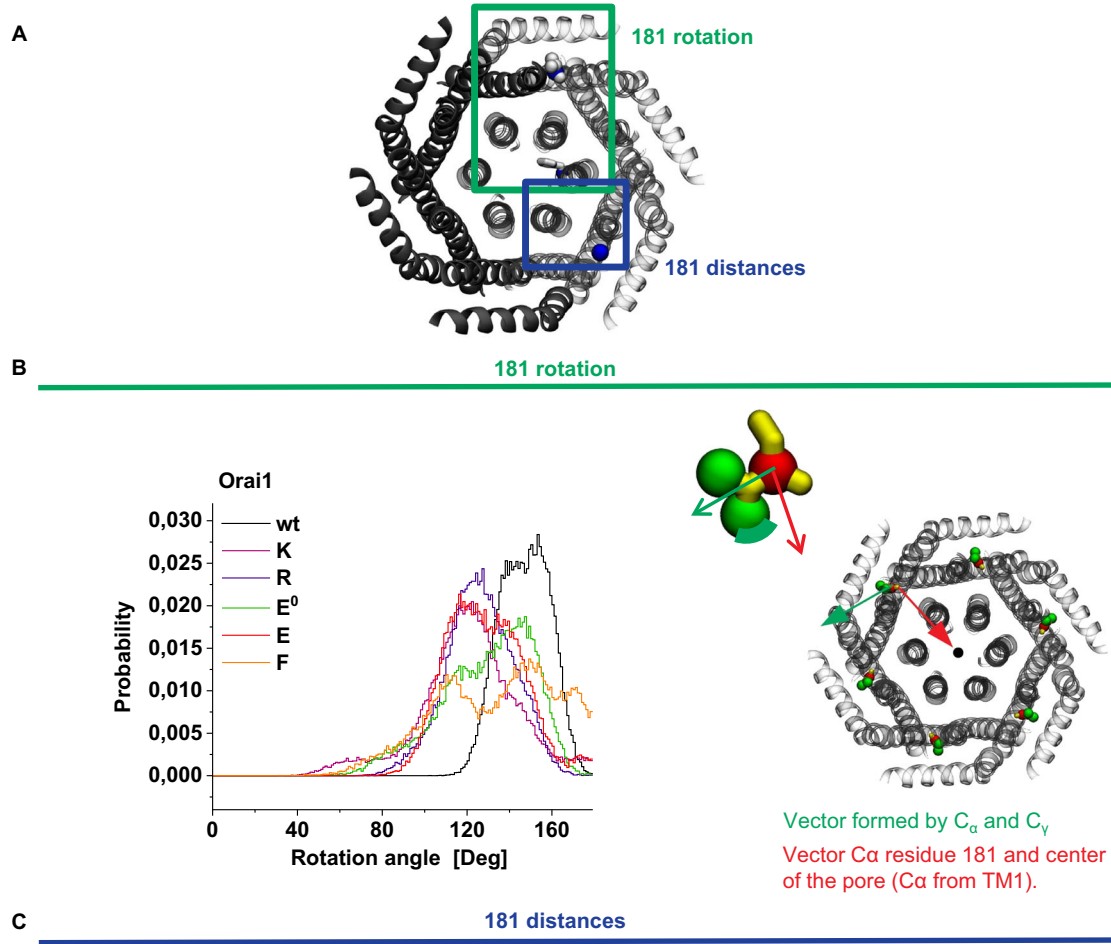

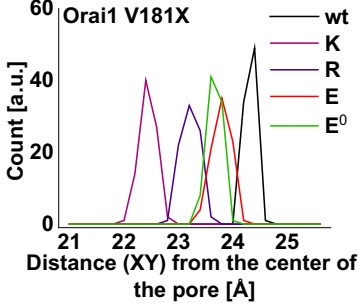

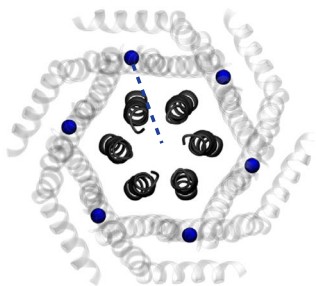

**Fig. 5 | Structural impact of V181X (X = K, R, E, E⁰, F) on peripheral TM-interfaces and the pore of Orai1. A** Bottom view of the wt Orai1 with TM1 (residues 80–110), TM2 (residues 120–150), TM3 (residues 170–195) and TM4 (residues 235–262) shown as ribbons. Half of the protein is shown as solid ribbons while the other half is depicted as transparent ribbons with the position of the analyzed residues highlighted. Green square: the position of V181 shown in gray beads with the Cα shown as a blue bead. Blue square: the Cα from V181 is shown as a blue bead. **B** Left: rotation of V181 is shown for the wild-type and its mutants. The rotation angle is defined by using the vector formed by the center of the pore (defined by the geometric center of the Cα from residues 80–110) and the Cα residue of residue 181 and the vector formed by the Cα and Cγ from residue 181. Plots for the mutants are using dashed lines and with the following color code, V181K (purple), V181R (blue), V181E (red), V181E⁰ (teal), V181F (orange) and wt (black). Data for wt Orai1 are represented by a solid black curve. Right: bottom view of the channel with the TM region of the protein represented as transparent ribbons. Cα and Cγ from residues 181 used for the directional green vector are depicted with red and green beads respectively. The directional vector toward the pore in red is calculated using the Cα from residues 181 and the center of the pore (gray beads) defined by the geometric center of the Cα from residues 80 to 110. **C** Left: distance from the pore center (*XY* plane) of the charged moieties at position 181 (ammonium, carboxyl and guanidinium for V181K, V181E and V181R, respectively). Data for V181K, V181R, V181E, V181E⁰ and wt are presented in purple, blue, red, teal and black, respectively. Right: bottom view of the TMs of Orai1 depicted as ghostly ribbons. The Cα from residues 181 are shown as blue beads. Direction toward the center of the pore is presented using blue dashed line.

protein parts, is fundamental to secondary and tertiary structure formation and enhancement of protein solubility[41–44]. Fluctuations in hydration water can affect protein dynamics and function. This is also an important concept for ion pores, in which wetting is an essential requirement for ion flow[45]. Hydration of the Orai1 pore is locally controlled by amino acid properties within the hydrophobic gate[23] and the basic region[27]. Furthermore, hydration of the backside of the TM1 helix contributes to the maintenance of the $Ca^{2+}$ selectivity of Orai1[28]. Here, we explored the role of hydration of the peripheral TM-interfaces on Orai1 pore hydration and ion permeability by investigating the activation process of different V181 mutations in TM3. We

**Fig. 6 | Small hydrophobic substitutions of Orai1 residues C143 and F253 located in the same membrane plane as V181K significantly reduce Orai1-V181K currents. A** Schematics showing side and top views of the TM3/TM4-interface along with TM2′ of the next subunit with the tested residues K181, C143 and F253. Time courses of Ca²⁺ current densities after whole-cell break-in of Orai1-V181K-F253A/W (**B**) and Orai1-C143A/F/L/W-V181K (**D**) compared to Orai1-V181K (indicated by /) in the absence of STIM1. **C, E** Block diagrams of maximal whole-cell current densities of mutants recorded in (**B, D**). For the presented bar diagrams, ANOVA (one-way) or Welch-ANOVA was employed for statistical analyses with differences considered statistically significant at $p < 0.05$ indicated with asterisks (see Supplementary Data 1).

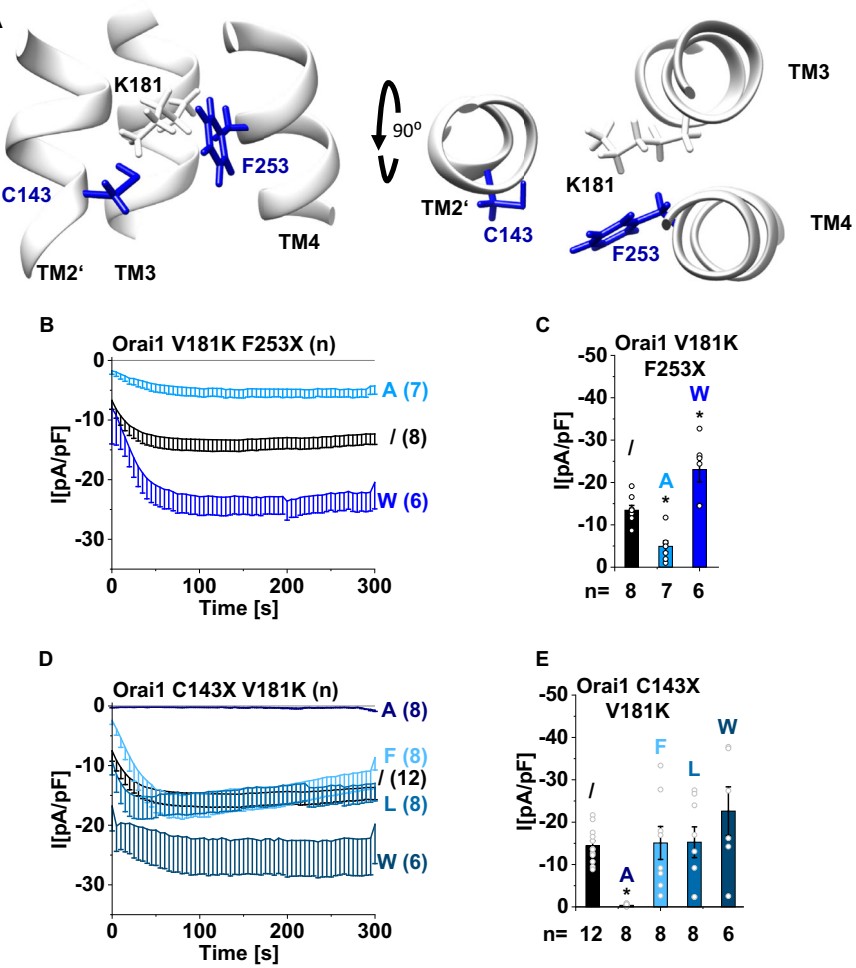

presented a substantial body of observations and arguments consistently indicating that increased hydration along the peripheral TM-interface of Orai1, as detected in silico, facilitates pore opening. This process is possibly mimicked by the physiological pore opening mechanism triggered by STIM1.

Our functional screen of Orai1-V181X mutants showed that in particular some charged substitutions (K/E/R) led to robust constitutive activity in the absence of STIM1. Interestingly, the V181D-substitution left the channel inactive, possibly due to decreased side chain size compared to E. Investigations in the presence of STIM1 revealed weak (Orai1-V181K) or no further current enhancements (Orai1-V181E/R). Orai1-V181D showed significantly reduced STIM1-induced activation compared to wt Orai1. The reduced or lacking responsiveness of these mutants with charged substitutions to STIM1 may underlie their altered coupling. Indeed, V181E/R-substitution in Orai1 led to delayed and slightly decreased FRET and V181D-mutation resulted in significantly reduced FRET with STIM1, while FRET of Orai1-V181K with STIM1 exhibited comparable behavior like wt Orai1. Nevertheless, Orai1-V181K/E/R showed reduced homomerization FRET, potentially due to changed conformational rearrangement of the Orai1 C-termini, which might be another reason why Orai1-V181K/E/R-mutants are not further activated by STIM1. Despite these altered responses to STIM1, the crucial piece of the work represents that a charge close to the center of TM3 or TM4 can induce CRAC channel-like activity already in the absence of STIM1, which motivated us to perform further investigations using MD simulations.

Among the electrophysiologically tested Orai1-V181X mutants, we examined those with the most pronounced and deviant effects (Orai1 V181K, R, E⁰, E, A, F) compared to wt Orai1 using MD simulations. We focused in particular on the effect of residues with a charged side chain at

position 181. Typically, the transfer of polar amino acids from an aqueous to a hydrophobic environment is energetically unfavorable. Indeed, the incorporation of basic amino acids (K, R) at other positions in Orai1 TM domains hampered PM expression (e.g. Orai1 L130R, Orai1 L193K). Remarkably, charged residues at several positions along the TM3/TM4-interface (e.g. V181, A254) preserved PM expression, and we observed constitutive activity for substitutions at V181 and A254. The latter is associated not only with enhanced pore hydration, but also with hydration along the peripheral TM-interfaces[28]. These charge-induced structural effects appear to be strong, since even a mutant (A254E) with partially restricted PM localization still exhibited robust constitutive current densities.

Our findings are consistent with previous MD reports showing that the presence of an electrostatic charge embedded in a low dielectric environment creates water fingers allowing the charge to connect to the higher dielectric environment associated with the bulk water[46]. This behavior reduces the energetic penalty of an electrostatic charge buried within a hydrophobic environment. This ability of charged residues has already been investigated previously in Orai1 along the basic region[27] as well as in the crevices located behind the conducting pore[28]. In the basic region, positive charges trigger wetting, which determines the extent of Ca²⁺ permeation[27]. In areas at the backside of the pore, glutamic acid (E190) in TM3 determines hydration, which together with a nearby lysine (K198) drives selectivity toward cations[28]. Similarly, hydration at the peripheral TM-interfaces may occur through the insertion of a charged residue. The formation of a local hydration shell can drag water molecules from the cytosolic pore mouth to the dry TM3/TM4-interface and vice versa achieving a dynamic water interchange toward previously dewetted protein parts. Additionally, the hydration shell will lead to an increase in side chain volume[47] possibly further dilating the peripheral TM-interface. This is supported by our

**Fig. 7 | Large hydrophobic residues in positions A177 and F257 located along the TM3/TM4-interface one helical turn closer to the cytosol than V181K decrease or abolish the constitutive activity. A** Schematics showing side and top views of the TM3/TM4-interface along with TM2′ of the next subunit with the tested residues K181, A177 and F257. Time courses of Ca²⁺ current densities after whole-cell break-in of Orai1-A177F/L/W-V181K (**B**) and Orai1-V181K-F257A/W (**D**) compared to Orai1-V181K in the absence of STIM1. **C, E** Block diagrams of maximal whole-cell current densities of mutants recorded in (**B**, **D**). Block diagram (**E**) additionally shows the whole-cell current densities of Orai1-A177W-V181K-F257W and Orai1-V181R-F257A/V/W compared to Orai1-V181K and Orai1-V181R (indicated by /), respectively. For the presented bar diagrams, ANOVA (one-way/Kruskal–Wallis ANOVA) or Welch-ANOVA was employed for statistical analyses with differences considered statistically significant at $p < 0.05$ indicated with asterisks (see Supplementary Data 1).

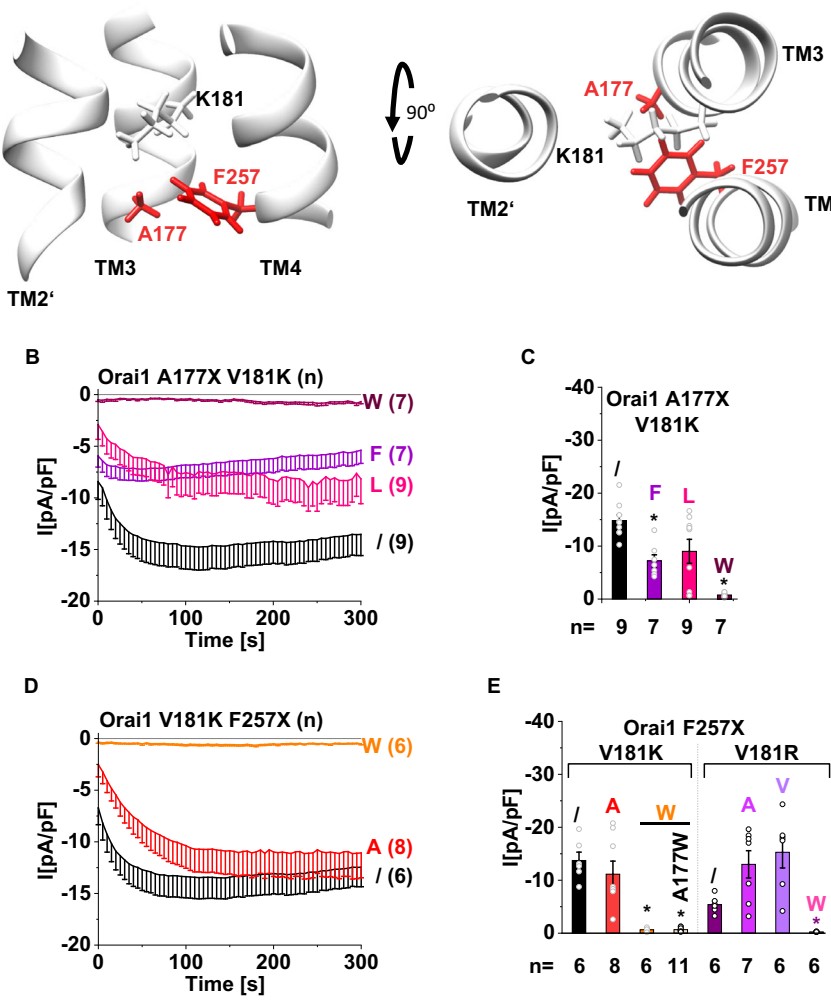

calculations that water dipoles interact preferentially with point charges and less with water and hydrophobic components (Supplementary Information, Supplementary Table 1), which is consistent with our observation that charged substitutions at V181 tend to increase the area formed by C143 in TM2, A177 in TM3 and F253 in TM4. In accord with our hypothesis that dilation of peripheral TM-interfaces is involved in pore opening a recent MD simulation study revealed that pore opening involves a breakdown of contacts between TM3 and TM4b as well as TM3 and TM2[48]. Interestingly, also the V181F-substitution tended to enhance the area formed by C143 in TM2, A177 in TM3 and F253 in TM4 likely due to its larger size. However, the mutant-channel remained inactive not only in the absence, but also in the presence of STIM1. This could arise from an improved water-shielding effect arising from the phenylalanine side chain maintaining a dewetted TM3/TM4-interface compared to wt Orai1. A possible reason for the weak constitutive activity after the substitution of even larger aromatic amino acids, namely Y or W, on V181, despite their water-shielding ability, could be due to a widening of the non-pore-lining TM interfaces caused by the size of the side chain. This conformational change at the periphery could propagate to the center of the channel complex to cause pore opening. Testing these hypotheses will be the subject of our future studies.

As previously described, wetting and de-wetting of the ion channel pore can trigger conformational transitions[45]. Similarly, our results demonstrate that hydration of the peripheral TM-interface is responsible for global conformational changes affecting the entire channel complex. On the one hand, we suggest a structural reorganization of the periphery of the channel complex, as the FRET of the C-terminally labeled constitutive Orai1-V181K/E/R was reduced compared to wt. This is possibly due to a hydration-induced restructuring of the C-termini. In agreement with this,

functional studies revealed that STIM1-mediated pore opening is associated with a conformational change along the Orai1 C-termini[49,50]. On the other hand, our simulations show that hydration along the peripheral TM-interface leads to a previously unobserved rotation of charged side chains at position 181 toward the pore, which is associated with enhanced hydration along the pore.

Interestingly, the different electrostatic mutations exhibit slight variations in the extent of rotation, the distance to the pore, the degree of hydration and the extent of constitutive activity, which can be explained by the distinct physicochemical properties of their side chains. Despite similar length and polarity, the guanidinium group of arginine shows a tendency for higher hydrophobicity than the amino part of the lysine, depending on the environment[51–53]. From our simulations, we explain these observations by the ability of R181 to form a cation–π interaction with the phenyl ring of F257. This interaction reduces the likelihood of the guanidinium moiety entering the water crevice in the back pores and may explain the reduced constitutive activity of Orai1-V181R as compared to Orai1-V181K. In comparison, both K181 and E181 were able to fully direct their side chains toward the water-rich clefts. The reduced constitutive activity of Orai1-V181E compared to Orai1-V181K could potentially be explained by the altered length and flexibility of the glutamate versus lysine side chain. This enables the lysine to orient its side chain directly toward the conducting pore and to bring its side chain amino group closer to it. Thus, it is able to recruit more water molecules around the hydrophobic gate.

Concerning pore hydration provoked by charged substitutions of V181, our results demonstrate that only V181K showed an increase in hydration and size of the hydrophobic gate and the basic region. In contrast, glutamate or arginine at position 181 increased only the hydration and size of the basic

region, but to a lower extent compared to that of the hydrophobic gate. This is likely related to the physicochemical property of the charged side chains at position 181. Hydrophobic or polar substitutions have no or comparatively weak effects on pore size or hydration of the basic region. A closer look at the charge by switching the protonation state of the glutamate suggests a combined effect of charge and side chain length on hydration and pore opening, which is due to enhanced hydration of V181E$^0$ compared to wt Orai1. In support of the role of the side chain, not only Lys (V181K) and Glu (V181E) but also the neutral Gln at position A254 (A254Q) triggered robust constitutive activity. For aromatic substitutions (F/Y/W) at position V181, leading to LoF or weak constitutive acitivity, the hydrophobic shielding seems to dominate the effect of side chain size. All constitutively active mutants had also a small impact on hydration and pore radius around the selectivity filter. The slight, but significant, change in $V_{rev}$ of constitutive mutants without STIM1, compared to with STIM1, may result from these effects on the selectivity filter combined with impacts on the pore's hydrophobic and basic regions due to a charged side chain at position 181.

Of all the Orai1-mutants tested in this study, the presence of lysine at position 181 or 254 resulted in the greatest constitutive activity of the channel. Due to the multiple effects of V181K on the channel complex, it represents the perfect mechanistic example of our earlier finding that a global conformational change is required for the Orai1-channel pore opening[16]. Since Orai1-V181K and Orai1-A254K currents are (i) strongly inward rectifying, (ii) still highly Ca$^{2+}$-selective and (iii) even higher than wt STIM1/Orai1-currents, these substitutions seem to abolish energetic cost otherwise provided upon STIM1-coupling to induce pore opening. Indeed, Orai1-V181K showed only slightly enhanced STIM1-induced activation despite unaffected store-dependent STIM1-coupling, while Orai1-A254K was incapable of binding to STIM1. The wetting-induced conformational flexibility of the peripheral TM-interfaces may be a key culprit for the widening of the TM3/TM4-interface finally triggering a wave of interdependent motions to induce pore opening, as described previously[16]. The observed effects of hydration-induced dilation of the peripheral TM-interfaces are supported by our finding that the combination of a charge with an aromatic side chain in the plane of V181 further increased Ca$^{2+}$ permeation. Conversely, shielding the charge by hydrophobic residues one helical turn closer to the cytosol (e.g. V181K F257W) reduced or abolished Ca$^{2+}$ currents. This highlights the role of efficient de-wetting driven by the hydrophobic effect along peripheral TM-interfaces in stabilizing the inactive state. Hence, the formation of a hydrophobic core along the TM3/TM4-interface as present in wt Orai1 provides thermodynamic stability maintaining the closed state.

Regarding the physiological situation, we hypothesize that STIM1-coupling to wt Orai1 may entail long-range C-terminal structural reorientations including TM4, possibly allowing water recruitment into the TM3/TM4-interface, where water would preferentially form hydrogen bonds with each other[44,54–56].

However, there is still enhancement in the constitutive activity and/or $V_{rev}$ of Orai1-V181K and Orai1-A254K in the presence of STIM1. This aligns with the biophysical properties of several GoF mutants, which only resemble those of the CRAC channel in the presence of STIM1[30,57]. Therefore, STIM1 fine-tunes Orai1 pore opening after initial coupling to the Orai1 C-terminal regions. Further investigation is required to determine whether this is exclusively achieved through the coupling of STIM1 to the C-terminus of Orai1, or if it requires the additional involvement of the loop2 and/or the N-terminus.

In conclusion, our study provides valuable insights into the molecular mechanisms that govern peripheral Orai1-channel gating, which can serve as a foundation for future investigations aimed at uncovering the precise inter-TM domain motions regulating Orai1 pore opening. Since TM4 (P245) is associated with Stormorken-like syndrome[13], one might expect similar, albeit not as pronounced, conformational rearrangements for this pathological mutant as induced by positive charges in TM3 or TM4. Thus, our studies offer a possible explanation for the opening mechanism of this disease-relevant mutation. Furthermore, our findings could form the basis

for improved drug design that modulates pore opening by targeting the peripheral TM-interfaces of Orai1.

## Methods
### Molecular biology
For N-terminal fluorescence labeling of human Orai1 (Orai1; accession number NM_032790, kindly provided by the laboratory of A. Rao), the constructs were cloned into the pEYFP-C1 (Clontech) expression vector via KpnI/XbaI (Orai1) restriction sites. Site-directed mutagenesis of all the mutants was performed using the QuikChange™ XL site-directed mutagenesis kit (Stratagene) with the corresponding Orai1 constructs serving as a template. Orai1-GCamp6f (purchased from Addgene[58]).

Human STIM1 (STIM1; Accession Number: NM_003156), N-terminally ECFP-tagged, was kindly provided by T. Meyer's Lab, Stanford University. The integrity of all resulting clones was confirmed by sequence analysis (Eurofins Genomics/Microsynth).

### Cell culture and transfection
Transient transfection of human embryonic kidney (HEK) 293 cells was performed using the TransFectin Lipid Reagent (Bio-Rad) (New England Biolabs)[59]. For each transfection, Orai1 plasmids together with STIM1 plasmids, R-Geco1.2 (purchased from Addgene[60]) or NFAT were used at a 1:1 ratio. Regularly, potential cell contamination with mycoplasma species was tested using VenorGem Advanced Mycoplasma Detection kit (VenorGEM)[61,62].

### Ca$^{2+}$ imaging
For Ca$^{2+}$ fluorescence measurements, HEK293 cells, transfected with Orai1/R-Geco1.2, were grown on coverslips for 1 day. Coverslips were transferred to an extracellular solution without Ca$^{2+}$ and mounted on an Axiovert 135 inverted microscope (Zeiss, Germany) equipped with a sCMOS-Panda digitale Scientific Grade camera 4.2 MPixel and a LedHUB LED Light-Engine light source. Excitation of R-Geco1.2 was obtained using the LED spanning 500–600 nm together with a Chroma filter allowing excitation between 540 and 580 nm and emission between 590 and 660 nm. Ca$^{2+}$ measurements are shown as normalized intensities of R-Geco1.2 fluorescence in HEK293 cells[63]. Image acquisition and intensity recordings were performed with Visiview5.0.0.0 software (Visitron Systems). A Thomas Wisa perfusion pump was used for extracellular solution exchange during the experiment[63]. All experiments were performed on 3 days at standard laboratory conditions using extracellular solutions containing (in mM): 140 NaCl, 10 HEPES, 10 glucose, 5 KCl, 1 MgCl$_2$, pH 7.4 and 0 or 2 CaCl$_2$, respectively.

### Electrophysiology
Expression patterns and levels of the various constructs were carefully monitored by fluorescence microscopy and were not significantly changed by the introduced mutations. Electrophysiological experiments were performed at 18–20 °C, using the patch-clamp technique in the whole-cell recording configuration. For Orai1, STIM1/Orai1 current measurements, voltage ramps were applied every 5 s from a holding potential of 0 mV, covering a range from −90 to +90 mV over 1 s. The internal pipette solution for passive store-depletion contained (in mM) 3.5 MgCl$_2$, 145 cesium methane sulfonate, 8 NaCl, 10 HEPES, 20 EGTA at pH 7.2. The extracellular solution consisted of (in mM) 145 NaCl, 5 CsCl, 1 MgCl$_2$, 10 HEPES, 10 glucose, 10 CaCl$_2$ at pH 7.4. Applied voltages were not corrected for the liquid junction potential, which was determined as +12 mV. All currents were leak-corrected by subtraction of the leak current which remained following 10 µM La$^{3+}$ application. All experiments were carried out at least on 3 different days. Bar graphs in the figures display maximum current densities for Orai1 proteins in the absence and presence of STIM1.

### Confocal FRET fluorescence microscopy
Confocal FRET microscopy was carried out at room temperature 18–24 h after transfection. The standard extracellular solution contained (in mM):

145 NaCl, 5 KCl, 10 HEPES, 10 glucose, 1 $MgCl_2$, 2 $CaCl_2$ and was set to pH 7.4. The experimental setup consisted of a CSU-X1 Real-Time Confocal System (Yokogawa Electric Corporation, Japan) combined with two CoolSNAP HQ2 CCD cameras (Photometrics, AZ, USA). The installation was also fitted with a dual port adapter (dichroic, 505lp; cyan emission filter, 470/24; yellow emission filter, 535/30; Chroma Technology Corporation, VT, USA). An Axio Observer.Z1 inverted microscope (Carl Zeiss, Oberkochen, Germany) and two diode lasers (445 and 515 nm, Visitron Systems, Puchheim, Germany) were connected to the described configuration. All described components were positioned on a Vision IsoStation antivibration table (Newport Corporation, CA, USA). Image recording and control of the confocal system were carried out with the VisiView software package (v.2.1.4, Visitron Systems). The illumination times for individual sets of images (CFP, YFP, FRET) that were recorded consecutively with a minimum delay were kept in a range of 100–300 ms. Due to cross-excitation and spectral bleed-through, image correction before any FRET calculation was required. YFP cross-excitation (a) and CFP crosstalk (b) calibration factors were therefore determined on each measurement day using separate samples in which cells only expressed CFP or YFP proteins. FRET analysis was limited to pixels with a CFP:YFP ratio between 0.1:10 and 10:0.1. After this threshold determination as well as background signal subtraction, the apparent FRET efficiency $E_{app}$ was calculated on a pixel-to-pixel basis. This was performed with a custom program integrated into MATLAB (v.7.11.0, The MathWorks, Inc., MA, USA) according to the following equation:

$$E_{app} = \frac{I_{FRET} - a\,I_{YFP} - b\,I_{CFP}}{I_{FRET} - aI_{YFP} + (G-b)I_{CFP}} \qquad (1)$$

where $I_{FRET}$, $I_{YFP}$ and $I_{CFP}$ denote the intensities of the FRET, YFP and CFP images, respectively. $G$ denotes a microscope-specific constant parameter (dimensionless) that was experimentally determined as 2.75 for our setup[64].

### NFAT-translocation studies using confocal fluorescence microscopy

Images of Orai1 and Orai1-mutants as well as CFP-NFAT (NFATc1[65]) localization were created and analyzed with a custom-made software integrated into MATLAB (v7.11.0, The MathWorks, Inc., MA, USA). ImageJ[66] was employed for subcellular localization analysis of the NFAT transcription factors by intensity measurements of the cytosol and nucleus, distinguishing between three different populations with different nucleus/cytosol ratios: inactive (<0.85), homogenous (0.85–1.15), and active (>1.15)[67].

NFAT measurements were performed 18–24 h after cell transfection and conducted in media containing 2 mM $Ca^{2+}$. For time-dependent NFAT-translocation, cells were transfected and grown in 0 mM $Ca^{2+}$ medium for 18–24 h. After this time the media was exchanged by 2 mM extracellular $Ca^{2+}$ solution and incubated for the indicated time points (2 and 4 h).

### Molecular dynamics simulations

A structural model of the human Orai1-channel was obtained using a homology modeling procedure[24] based on the *Drosophila melanogaster* Orai structure (Protein Data Bank ID: 4HKR)[9]. Using this Orai1 structure, mutants were created by in silico point mutation in each subunit using the CHARMM-GUI web interface[68]. The membrane is constituted of 394 POPC (1-palmitoyl-2-oleoyl-glycero-3-phosphocholine) molecules. The upper leaflet is made of 206 lipids while the lower leaflet is composed of 188 lipids for an initial box size of 131.104 Å × 131.104 Å × 130.89 Å. An ionic strength corresponding to 0.10 M of $CaCl_2$ was used for simulations of systems with (initially) closed channel complexes. The minimization and equilibration steps from CHARMM-GUI were kept[68]. Production was then run for 400 ns or even 550 ns (for Orai1-V181E) for each system with three independent replicas. For all the simulations, CHARMM36 force field for proteins[69] and lipids[70] was used while the TIP3P model was used for water molecules[71]. Charges and van der Waals parameters for the ions are derived

by using a rescaling scheme and a mean-field approach[72]. All titratable amino acids, except for the glutamic acid at position 181 where both protonation states were investigated, are set to their standard protonation state at pH 7.

Molecular dynamics (MD) simulations were all performed with GROMACS (version 2020)[73]. A leap-frog algorithm for integrating Newton's equations of motion was used with a time step of 2 fs. The LINCS algorithm was used to constrain all hydrogen bonds[74]. The pressure was kept constant at 1.0 bar by using the Parrinello–Rahman barostat with a coupling constant of 5.0 ps and an isothermal compressibility of $4.5 \times 10^{-5}$ $bar^{-1}$[75]. A semi-isotropic scheme was used where the pressure in the xy directions is decoupled from the direction normal to the bilayer (z-direction). The Nosé–Hoover thermostat was used to keep the temperature constant at 310.0 K with a coupling constant of 1.0 ps[76,77]. The water with the ions, membrane and proteins were coupled to separate thermostats. Long-range electrostatics were treated with the particle mesh Ewald scheme with a real-space cutoff set at 1.2 nm[78,79]. The cutoff for Lennard-Jones interactions was set to 1.2 nm.

For water molecule analysis, only the water molecules contained within a cylinder of 10 Å around the center of the pore of the Orai channel were considered for the main conducting pore while a cylinder of 30 Å was used when analyzing the back pore hydration as well.

The rotation angles associated with residue 181 were calculated by using the vector formed by the center of the pore (defined by the geometric center of the Cα from residues 80–110) and the Cα residue of residue 181 and the vector formed by the Cα and Cγ from residue 181.

The distances for residue 181 are taken as the distance between the geometric center of the pore defined as the Cα from residues 80–110 and the geometric center of the charge-carrying moiety (nitrogen atom for lysine, carbon and oxygen atom from the carboxylic acid moieties or guanidinium in the case of arginine).

RMSDs were calculated by fitting the Cα from the residues in TM1 (residues 80–110), TM2 (residues 120–150), TM3 (residues 179–185) and TM4 (235–262) to the crystal-based homology structure. All analyses have been performed on all three replicas.

The VMD 1.9.4 software[80] was used for rendering and analysis was performed with VMD and MDanalysis[81,82]. The HOLE[83] software was used to calculate the cavity size of the pore.

### Hydrophobicity profiles

Hydrophobicity profiles[34] were determined based on Roseman scale with a window size within the range of 9–25 amino acids.

### Statistics and reproducibility

Results are presented as means ± S.E.M. calculated for the indicated number of experiments. The Kruskal–Wallis ANOVA test was performed for statistical comparison of two independent samples considering differences statistically significant at $p < 0.01$ with Dunns post hoc test. For multiple independent samples, we tested for variance homogeneity by the Levene test. If variance homogeneity was not fulfilled, we performed instead of the ANOVA test, the Welch-ANOVA test. Subsequent to Welch-ANOVA we performed the Games–Howell post hoc test to determine the pairs which differ statistically significant ($p < 0.05$). Kolmogorov–Smirnov test was applied to prove the normal distribution of the respective datasets.

Experiments that assessed the characteristics of 3–4 constructs were carried out in paired comparison on the same day and repeated three times.

### Reporting summary

Further information on research design is available in the Nature Portfolio Reporting Summary linked to this article.

### Data availability

MD simulations were performed using a modeled human Orai1 structure based on 4HKR. The source data underlying Figs. 1–7, and Supplementary Figs. 1–10 and all sequencing data are provided as a Source Data file and are

deposited in an open public repository [https://doi.org/10.5281/zenodo.13944768]. Source data are provided with this paper.

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

## Acknowledgements

We thank S. Buchegger for excellent technical assistance. This research was funded by the Austrian Science Fund (FWF) projects [https://doi.org/10.55776/P32851], [https://doi.org/10.55776/P35900] and [https://doi.org/10.55776/P36202] to I.D. and [https://doi.org/10.55776/PAT6871323] to A.T. For open access purposes, the author has applied a CC BY public copyright license to any author-accepted manuscript version arising from this submission.

## Author contributions

V.H., A.T., D.B. and I.D. conceived and coordinated the study and wrote the paper. V.H., A.T., Ca.H., H.N. and S.W. performed and analyzed electrophysiological experiments. M.Fr. performed and analyzed $Ca^{2+}$ imaging experiments. A.W., M.S., H.G. and M.P. carried out fluorescence microscopy experiments. V.H., A.T., A.W., J.S., Y.N., S.H., N.K., M.F., M.Fr. and C.H. contributed to molecular biology and biochemistry. D.B. performed and supervised MD simulations. H.K. performed calculations of water interactions. All authors reviewed the results and approved the final version of the manuscript.

## Competing interests

The authors declare no competing interests.

## Consent to publish

All authors have read and approved its submission to this journal.
