## [Transparent Peer Review file · Communications Biology]

Water in peripheral TM-interfaces of Orai1 channels triggers pore opening

Corresponding Author: Professor Isabella Derler

Version 0:

Reviewer comments:

Reviewer #1

(Remarks to the Author)

I have reviewed the manuscript entitled "Water in peripheral TM-interfaces of Orai1 channels triggers pore opening" by Valentina Hopf et al. In this study the authors have examined the structural rearrangements in the pore of the Ca²⁺ channel Orai1 when it is activated by the physiological activator stromal interaction molecule 1 (STIM1). Previous studies by this group and others have shown that a series of structural rearrangements occur within the entire channel complex involving a series of gating checkpoints. By focusing on the gating mechanism operating along the peripheral transmembrane (TM) 3/TM4-interface, they here they report that charge substitutions close to the center of TM3 or TM4 lead to constitutively active Orai1 variants triggering nuclear factor of activated T-cell (NFAT) translocation into the nucleus. Molecular dynamics simulations unveil that this gain-of-function correlates with enhanced hydration at peripheral TM-interfaces, leading to local swelling of the channel periphery and global conformational changes permitting pore opening. The findings indicate that efficient dehydration of the peripheral TM-interfaces driven by the hydrophobic effect is critical for maintaining the closed state of Orai1. They propose a physiological gating mechanism involving concomitant hydration and widening of peripheral TM interfaces after STIM1 binding to trigger Orai1 opening.

Overall, this is an important study that further reveals important molecular and structural understanding of the pore of the Orai1 channel as well as the regulation of its function.

I have several points that will need to be addressed.

1. The colors that have been picked for the models in the paper make it very difficult to see details. Also, many figures include overlapping data from several mutants – it is very difficult to resolve the data for each. So I would suggest that data for WT as well as the important mutants highlighted should be shown individually.
2. NFAT activation by the mutants has been noted as one of the major features of these mutants. However, there is very little Ca²⁺ data to support whether local Ca²⁺ in these mutants are similar. Also, previously it has been shown that Orai1 has to be recruited into ER/PM junctions by STIM proteins in order for them to be coupled with NFAT activation. So, in that context, the weak activation of NFAT by some of the mutants and restoration of function in presence of STIM1 could be an effect of recruitment into junctions and not just loss of channel function. In that context, NFAT recruitment (both rates and final) need to be recorded and quantitated in the presence and absence of STIM1 (and with store depletion). It appears that all the experiments shown do not have any store depleting conditions. I urge the authors to take a closer look at this.
3. The paper would benefit from discussion as to why increasing channel pore size- did not cause changes in selectivity.
4. Finally the paper will benefit from extensive editing and proofing.

Minor points:

Fig. 1:

- i. Add WT data (currents) either within panels or as a separate panel.
- ii. What does "-" above first bar in graph refer to – WT Orai1 ?

Supple Fig 1

- i. Clarify how some mutants give robust NFAT translocation but poor function. Constitutively active Orai1 might not be coupled with AKAP/calcineurin and so cannot trigger NFAT activation. Need to repeat with STIM1 and store depletion.
- ii. Fig J is wrongly referred to in the text. Fig does not show WT or the data is obscured due to color ?

Supple Table 2 is missing

Fig 2.

i. No WT shown

ii. How is NFAT translocation in K, E and Q mutants same when function is slower in the latter two. There are no Ca²⁺ data for these mutants, making it difficult to assess what is going on.

Fig.3

i. Too many traces shown together. There is some need to quantitate these data ? (in supple Fig 3 J- WT data is buried under everything else and cannot be seen). These and other similar figures with multiple traces etc. need to be replotted so the main message can get across to the reader.

ii. Describe the images in Supplemental fig A and B – make the Ca²⁺ binding pocket more clearly visible- difficult to differentiate between yellow and white.

iii. It might be useful to move Supple. Fig 3 A, B and J-L into main Figure 3. These data support the data in the main figure. Figure 4 and later figures – please consider changing the colors on the models and making data clearly seen in the various experimental groups.

Reviewer #2

(Remarks to the Author)

This paper by Hopf and colleagues aims to understand the mechanistic basis of gain-of-function (GOF) effects of a previously identified mutant, V181K (Tiffner et al, 2020, 2021). Introduction of some charged mutations at V181 (e.g., V181K/E/R) cause constitutive channel activity, and in the previous study, it was concluded that V181 is part of a network of “checkpoints” in Orai1 that controls channel gating. In the current study, the main advance here is addition of MD simulations showing that the water content around V181 increases with the V181K and V181E mutations and decreases with the loss-of-function V181F mutation. Additional analysis of the effects of mutating residues near V181 are also shown and inferences are made regarding putative functional interactions between V181 and these other residues based on alterations in current magnitudes with the additional mutations.

Although the manuscript has a lot of data and the idea of water in membrane proteins embedded within lipids is potentially interesting, I find the study to be mainly correlative, qualitative, and the evidence for increased water around the TM3/4 helices is based only on MD simulations with some critical theoretical assumptions. Further, although the potential biophysical mechanisms for why water increases around V181 with charged mutations is explained in the Discussion, the relevance of this phenomenon for gating/activation of the WT channel with the native Val remains unclear. Thus, the significance of the study appears more relevant to understanding the basis of the GOF phenotype of the mutant channel rather than improved understanding of the activation of the WT channel. In other words, what is the physiological relevance of water at the TM3/4 interface for the native channel?

Main points

1) Figure 1E. It is important to show whether the various Orai1 V181X mutants bind to STIM1. Orai1+STIM1 currents don't change amplitude in most of the mutants. Is reduced current in some of the V181 mutants (in presence of STIM1) due to reduced STIM1 binding ? or something else?

2) Likewise, in Figure 2, a lack of change in current magnitudes in the presence of STIM1 (relative to without STIM1) suggests that STIM1 is not causing additional activation of the GOF Orai1 mutant. This is interesting and needs to be better understood. If water around V181 (in V181E for e.g.) is having a detrimental effect on STIM1-induced Orai gating, this suggests that the water at the TM3-4 interface may impede physiological (STIM1) activation of Orai1.

3) P 9 and Figure S2F: Much is made of the change in FRET between the CFP and YFP tags at the C-terminal ends of Orai1. Specifically, a difference in FRET between the V181 mutants is interpreted as decrease in Orai1 “homomerization”. This interpretation is highly problematic for numerous reasons. The individual channels will have unknown and variable numbers of CFP- and YFP-tags that cannot be controlled within the channel population in each cell. The FRET measurement here is a vague and nebulous measure of some combined effect of conformation, orientation, and multimerization of the probes expressed at unknown ratios and the underlying phenomenon contributing to the FRET difference cannot be untangled. This data is not convincing and should be better substantiated or removed.

4) Figure 4. I am concerned about the quality of the simulations. Many of the helices in the Figures shown seem to be floppy, with broken helicity, some seem to be unravelling, and the channel seems to show a lot of asymmetry and is not well-behaved in these snapshot images. What does a movie look like? If the helices are unstable and unraveling, this could produce inaccurate metrics for water content which could lead incorrect conclusions. E.g., even in panel for the WT, the TM1 helix seems to be unravelling.

5) There seems to be an unusually large amount of water within the TMs even in the WT channel, surprising for membrane proteins embedded in lipid. Obviously, this would depend on the simulation conditions, but a comparison to previous simulations of E190 mutations that group carried out in the Tiffner 2020 study might help provide some controls.

6) Figures 6. I understand the motivation for undertaking analysis of the double mutants at C143 and F253 with V181. The effects of these mutations are interpreted solely in the context of what they might do specifically to the GOF phenotype of the V181K mutant. However, it is equally likely that the reductions in V181K currents seen with hydrophobic substitutions are general in nature, and also effect the activation/gating of unrelated Orai1 constitutively active Orai1 mutants such as H206A

or V102X. This should be tested to provide stronger support for the proposed structural and functional interaction between C143/F253 with V181K.

7) Figure 7. Same issue as above. Are these generalized effects on channel gating independent of STIM1? Supplementary Figure 7G-H show that the A177X and F257X mutations inhibit currents of single mutants (even in the absence of V181K) by STIM1, suggesting that these are generalized effects on channel gating. Hence, the significance of these loss of these inhibitory mutations for the gating induced by V181K cannot be easily inferred.

Minor:

- Supplementary Figure 1E, I: The percentage of nuclear NFAT is shown as ~80% for V181G in panel E but is only about 30-40% in panel I. Please check and clarify.

- Figure 2. The statistics are vague and exactly what the “**” mean is unclear. Which groups are being compared here?

- Figure 3. The amino acids should be clearly indicated at the various axial positions. It appears from the hydration plots that the primary areas of increase hydration are far into the TM1 inner segment, around residues R83 or even further away from the membrane. How is this relevant to channel gating and relief of the free energy barrier in the pore?

Reviewer #3

(Remarks to the Author)

The manuscript by Hopf et al. used a combination of MD simulations and experimental assays to study the gating mechanism of Orai1 channels. This paper is well written and it represents a step forward in understanding the molecular basis of such mechanism. I can recommend the publication of the manuscript after the following points will be addressed:

1) The PDB code of the structure used for the computational part should be added in the first sentence of the MD simulations section to facilitate the readability of the text;

2) Which structure was used to produce the mutants? I suppose the structure of the wild type at the end of the dynamics but this should be clarified. Moreover, are the mutations introduced in all the subunits?

3) The number of the POPC molecules and the dimension of the simulation box should be provided;

Is there a specific reason why the authors used the TIP3P model which is different to that used in Frischauf et al? It is known from the literature that the water models can affect the results, especially if they play a key role in the study (see <https://pubs.acs.org/doi/10.1021/acsnano.0c04387>). Thus, the authors should comment on that;

4) The authors should specify what method or program they used to predict the protonation state of the residues;

5) What are the trajectories used to compute the RMSD profiles shown in Fig. S3? Is a plateau reached in all the replicas?

6) What are the structures used to compute the radius profiles with the HOLE program? Did they refer to the profile computed on the structures at the end of their dynamics or are they averaged over the 50 ns used for the post-processing analyses?

7) In the same vein, is the number of water molecules computed in a single frame at the end of the dynamics for the three replicas or do the profiles shown in Fig. 4C refer to the mean value over the overall last 50 ns and then averaged for the three replicas?

8) Page 10: when the authors state “In contrast, the mutants did not show the same level of structural stability, with RMSD values still increasing at 250 ns”, do they believe that these results are reliable even if the mutants did not reach their stability? Wouldn't it be more prudent to extend these simulations, especially in the systems V181F where the curves of TM3 and TM4 are still increasing?

9) It's not clear the statement “A deeper investigation of the water crevice around V181 revealed the involvement of residues C143 in TM2, A177 in TM3 and F253 in TM4”. Did the authors do a quantitative analysis to characterize their role or did they observe only the structure to deduce that? One can argue that also neighboring residues can play a role in that.

10) Page 11: the sentence “...the area formed by these three positions tended to increase concomitantly with the amount of water (Figure S3K,L)” seems to be slightly an overstatement if the error bars in panel L are considered.

11) Again in Fig. 5 (and also in S5) it's not clear which data have been used to compute the angles and the distances. Do they refer to a single subunit? Are they computed on a single trajectory or on all the replicas? Are they computed at the end of the dynamics or in the last 50 ns?

12) Page 14: the authors should provide a profile of the interaction R181-F257 during the simulations to state “In our simulations, this difference was evident in the ability of R181 to form a cation- π interaction with the phenyl ring of F257.”

13) Can the authors comment on their results in the view of <https://doi.org/10.3389/fmolb.2021.773388>?

Version 1:

Reviewer comments:

Reviewer #2

(Remarks to the Author)

The revised manuscript addresses some of the concerns raised in the previous round, but continues to suffer from major concerning problems of interpretation and claims that are not supported by the results.

The paper is also very difficult to understand and in many cases, the results presented are not interpreted at all.

The abstract concludes with a bold statement that “We propose a gating mechanism involving widening of peripheral TM-interfaces which may facilitate enhanced hydration after STIM1 binding to trigger Orai1 opening”. There is no data in the

paper supporting this sweeping statement. On the contrary, the data in Figure 1E clearly show that the mutants (E, R, Q, K), that evoke water at the TM3-TM4 interface are unresponsiveness to STIM1 binding. The D mutant in fact does not even bind STIM1 adequately (Supplem Figure 3). This statement is not supported by the data shown and must be changed.

The abstract also makes the claim that the MD simulations show that the V181X gain-of-function mutants show local swelling at the channel periphery. I do not see a correlation between the "local swelling", channel activity, and water anywhere. Crucially, the F mutant which shows more area between the 143, 177 and 253 is not even active (and hence negates the point of this abstract statement). The study should carry out analysis for the other mutants such as Y, L or S mutants which are active with STIM1 (but not w/o STIM1) to support the central conclusion of the paper.

Although the paper contains a large amount of data, much of it is very difficult to follow and in most cases, the crucial pieces of information are not even interpreted by the authors. For example, what does it mean that STIM1 does not activate the V181 gain-of-function mutants? Or that in some cases (D, E), they do not effectively bind to STIM1? The lack of gating by STIM1 is simply ignored.

One more crucial point: A careful examination of the current amplitudes in the absence (or even in the presence!) of STIM1 shown in Figure 1E and the amount of water in the TM3-TM4 interface shown in Figure 4D shows basically no correlation between the amount of water and the current magnitude. For example, the E mutation shows the maximal water, but this mutant is considerably less active than the K mutant which shows much less current. The F mutation shows some degree of water but is associated with virtually no current. The amount of water in the Ala mutant is about the same or less than the WT protein but this mutation is does show some constitutive current. Basically, I do not see the functional correlation between the amount of water in the TM3-TM4 interface and channel activity. This undercuts the major premise and key conclusion of the paper.

Finally, the positions of the TM1 residues along the y axis needs to be indicated in Figure 4C. This was requested in the initial review as well. This is essential for understanding the relative amounts of water along the pore.

Reviewer #3

(Remarks to the Author)

The authors addressed all my previous comments. I appreciated that they extended the simulations and added new analyses which improved the manuscript. I can therefore recommend its publication.

Reviewers' comments:

Reviewer #1 (Remarks to the Author):

I have reviewed the manuscript entitled "Water in peripheral TM-interfaces of Orai1 channels triggers pore opening" by Valentina Hopf et al. In this study the authors have examined the structural rearrangements in the pore of the Ca²⁺ channel Orai1 when it is activated by the physiological activator stromal interaction molecule 1 (STIM1). Previous studies by this group and others have shown that a series of structural rearrangements occur within the entire channel complex involving a series of gating checkpoints. By focusing on the gating mechanism operating along the peripheral transmembrane (TM) 3/TM4-interface, they here they report that charge substitutions close to the center of TM3 or TM4 lead to constitutively active Orai1 variants triggering nuclear factor of activated T-cell (NFAT) translocation into the nucleus. Molecular dynamics simulations unveil that this gain-of-function correlates with enhanced hydration at peripheral TM-interfaces, leading to local swelling of the channel periphery and global conformational changes permitting pore opening. The findings indicate that efficient dehydration of the peripheral TM-interfaces driven by the hydrophobic effect is critical for maintaining the closed state of Orai1. They propose a physiological gating mechanism involving concomitant hydration and widening of peripheral TM interfaces after STIM1 binding to trigger Orai1 opening.

Overall, this is an important study that further reveals important molecular and structural understanding of the pore of the Orai1 channel as well as the regulation of its function. I have several points that will need to be addressed.

We thank the reviewer for the constructive comments and positive evaluation of the manuscript.

1. The colors that have been picked for the models in the paper make it very difficult to see details. Also, many figures include overlapping data from several mutants – it is very difficult to resolve the data for each. So I would suggest that data for WT as well as the important mutants highlighted should be shown individually.

We have adapted the colors of the Orai1 models (Supplementary Fig. 6) and accordingly, the colors in the other figures/diagrams were adapted. In addition, we show the pore radius diagrams in Fig. 3 and the hydration diagrams in Fig. 3 and Fig. 4 separately for groups of charged, small polar and hydrophobic amino acids for better visualization. Otherwise for all time courses corresponding bar diagrams are shown for clarity.

2. NFAT activation by the mutants has been noted as one of the major features of these mutants. However, there is very little Ca²⁺ data to support whether local Ca²⁺ in these mutants are similar. Also, previously it has been shown that Orai1 has to be recruited into ER/PM junctions by STIM proteins in order for them to be coupled with NFAT activation. So, in that context, the weak activation of NFAT by some of the mutants and restoration of function in presence of STIM1 could be an effect of recruitment into junctions and not just loss of channel function. In that context, NFAT recruitment (both rates and final) need to be recorded and quantitated in the presence and absence of STIM1 (and with store depletion). It appears that all the experiments shown do not have any store depleting conditions. I urge the authors to take a closer look at this.

We analyzed the global Ca²⁺ increases using the most critical Orai1 mutants after co-expression with R-Geco1.2. We detected the highest Ca²⁺ levels for Orai1-V181K and slightly reduced (10-25%) Ca²⁺ levels for Orai1-V181E/R/G. A significant reduction of Ca²⁺ levels by more than 60% compared to Orai1-V181K was observed for Orai1-V181A/Y, while no constitutive elevations of Ca²⁺ levels were observed for Orai1-V181L as well as Orai1 (Supplementary Fig.1A-C). In contrast to the 2-3-fold smaller Ca²⁺ currents of Orai1-V181R/E versus Orai1-V181K, the Ca²⁺ levels of the 3 mutants did not differ that drastically. Furthermore, Orai1-V181G showed relatively higher Ca²⁺ levels than Ca²⁺ currents compared to Orai1-V181K. To clarify these differences, we performed additional experiments to

investigate local Ca²⁺ elevations using an Orai1 construct and corresponding mutants linked to the Ca²⁺ indicator (GCamp6f) (Supplementary Fig. 1D-F). Applying stepwise increases in extracellular Ca²⁺ concentration, we were able to determine a stepwise increase in cellular Ca²⁺-concentration for Orai1-V181K/R/E/A/G/Y-GCamp6f expressing cells, the magnitude and relative changes of which were in particular at lower extracellular Ca²⁺ concentrations more consistent with detected Ca²⁺-currents of the respective mutants (K<R<E<A=G=Y) (Supplementary Fig. 1D-F). Nevertheless, the still limited correlation of the relative differences between Ca²⁺ current levels versus Ca²⁺ concentration levels and NFAT translocation of Orai1-V181K/E/R mutant expressing cells could potentially be attributed to slightly distinct treatment of cells for Ca²⁺ imaging versus electrophysiology (cells for electrophysiology are reseeded 6-10 hours before the experiment) and/or to a lower variability of the tested cells using patch-clamp compared to fluorescence microscopy. The latter is likely due to the lower seal stability of cells with certain properties (e.g. higher transfected ones).

We agree that the weak activation of NFAT by some mutants does not necessarily mean that they are loss-of-function mutations. Indeed, they could be activated by STIM1, except for Orai1-V181F (Supplementary Fig. 1G-I). As recommended, we performed NFAT translocation experiments in the presence of STIM1 and upon store-depletion (Supplementary Fig. 2H-J) compared to the absence of STIM1 (Supplementary Fig. 2E-G). Most mutants, which are only weakly active in the absence of STIM1, showed an increase in NFAT translocation in the presence of STIM1 and upon store-depletion to comparable levels. Hence, the weak effect of some mutants in the absence of STIM1 is neither due to defect recruitment into ER/PM-junctions nor just loss of channel function.

3. The paper would benefit from discussion as to why increasing channel pore size- did not cause changes in selectivity.

While in the constitutively active Orai1-V181K/R/E mutants the pore diameter was drastically enhanced in the basic and hydrophobic region, the selectivity filter was slightly affected. The I/V relationship revealed a robust inward rectifying current, however, with the reversal potential of the constitutive Orai1-V181K/R/E mutants significantly leftward shifted by 10mV. Hence, we conclude that the increasing pore diameter in the selectivity filter is likely the reason for slightly perturbed selectivity. This aspect is addressed in the discussion as follows: "All constitutively active mutants had also a small impact on hydration and pore radius around the selectivity filter. The slight, but significant, change in V_{rev} of constitutive mutants without STIM1, compared to those with STIM1, may result from these effects on the selectivity filter combined with impacts on the pore's hydrophobic and basic regions due to a charged side chain at position 181 or 254." (p. 16, l. 627-630)

4. Finally the paper will benefit from extensive editing and proofing.

We carefully proofread and edited the manuscript.

Minor points:

Fig. 1:

- i. Add WT data (currents) either within panels or as a separate panel.
- ii. What does "-" above first bar in graph refer to – WT Orai1 ?

Supple Fig 1

The corresponding wild-type data were already added (e.g. Fig. 1 gray hatched bars were labeled with '-' in the previously submitted version). To make the labeling clearer, we exchanged the '-' by wt, thus, indicating wild-type (or in other words Orai1). In cases where a mutant is used as control, we exchanged the '-' by '/' and indicated it the figure legend (Fig. 6 & 7; Supplementary Fig. 8, 9 & 10).

- i. Clarify how some mutants give robust NFAT translocation but poor function. Constitutively active Orai1 might not be coupled with AKAP/calcineurin and so cannot trigger NFAT activation. Need to repeat with STIM1 and store depletion.

Since we observed for some mutants robust NFAT translocation, but weak function, we performed time-dependent NFAT translocation experiments in which cells were incubated in 0mM Ca²⁺-containing medium prior to the experiment. Solution exchange of a 0mM Ca²⁺- with a 2mM Ca²⁺-containing buffer was performed at the beginning of the experiment and NFAT translocation was monitored over 4 hours (Supplementary Fig. 2E-J). Indeed, we discovered more consistent NFAT translocation levels and currents levels. Hence, we concluded that in some cases prolonged exposure of the cells to Ca²⁺-containing media could be the reason for the drastically increased NFAT translocation.

Only for Orai1-V181K & -V181E/R the drastic differences in currents could not be seen in time-dependent NFAT translocation experiments. Hence, we additionally tried steady state experiments comparing Orai1-V181K/E/A/G-mediated NFAT translocation after keeping the cells in 0,1mM Ca²⁺-containing medium after transfection. Indeed, we discovered a tendency for lower NFAT translocation of Orai1-V181E versus Orai1-V181K expressing cells, while it was almost completely abolished for Orai1-V181A/G (Fig. 1, see below). Nevertheless, it seems that a low Ca²⁺ concentration is already sufficient for maximal NFAT translocation. Additionally, a slightly different treatment of cells in patch-clamp compared to fluorescence microscopy imaging could be an additional reason for difference in function and NFAT translocation as described above.

Furthermore, NFAT translocation experiments for most critical mutants were also performed in the presence of STIM1. All mutants showed high NFAT translocation in the presence of STIM1 and upon store-depletion comparable to wild-type conditions (Fig. S3H-J).

Figure 1: NFAT translocation of Orai1 mutants after incubation in 0,1mM Ca²⁺ after transfection. The average number of HEK293 cells that exhibit NFAT localization to the nucleus determined upon co-expression (CFP-NFAT) with YFP-Orai1 (wt), or Orai1-V181A/E/G/K after 8 h in 0,1 mM Ca²⁺-containing media.

ii. Fig J is wrongly referred to in the text. Fig does not show WT or the data is obscured due to color ?

This is adapted.

Supple Table 2 is missing

Suppl. Table 2 is a separate file containing all statistics results.

Fig 2.

i. No WT shown

WT was shown in gray hatched bars indicated with a '-' in the previously submitted version. The '-' has been exchanged by wt as stated above.

ii. How is NFAT translocation in K, E and Q mutants same when function is lower in the latter two. There are no Ca²⁺ data for these mutants, making it difficult to assess what is going on.

The experiments performed for the previously submitted version show NFAT translocation in a steady state.

We performed additional Ca²⁺ imaging experiments to examine the global Ca²⁺ entry of the Orai1-V181K/E/R mutants. While Orai1-V181K showed highest constitutive Ca²⁺ levels, Orai1-V181E/R exhibited 10-25% reduced constitutive Ca²⁺ entry (Supplementary Fig. 1A-C). Since the effects in Ca²⁺ imaging experiments were less pronounced than in electrophysiological studies, we performed additional studies on local Ca²⁺ entry with the mutants bound to GCaMP6f. Applying stepwise increases in extracellular Ca²⁺ concentration, we were able to determine a stepwise increase in cellular Ca²⁺-concentration for Orai1-V181K/R/E/A/G/Y-GCamp6f expressing cells, the magnitude and relative changes of which were in particular at lower extracellular Ca²⁺ concentrations more consistent with detected Ca²⁺-currents of the respective mutants (K<R<E<A=G=Y) (Supplementary Fig. 1D-F).

As described above, to understand the reason for less drastic changes of NFAT-translocation in Orai1-V181K & -V181E/R expressing cells compared to currents, we additionally tried steady state experiments comparing Orai1-V181K/E/A/G-mediated NFAT translocation after keeping the cells in 0,1mM Ca²⁺-containing medium after transfection. Indeed, we discovered a tendency for lower NFAT translocation of Orai1-V181E versus Orai1-V181K expressing cells, while it was almost completely abolished for Orai1-V181A/G (Fig. 1, see below). Nevertheless, it seems that a low Ca²⁺ concentration is already sufficient for maximal NFAT translocation. (Supplementary Fig. 1D-F).

Fig.3

i. Too many traces shown together. There is some need to quantitate these data ? (in suppl Fig 3 J- WT data is buried under everything else and cannot be seen). These and other similar figures with multiple traces etc. need to be replotted so the main message can get across to the reader.

As described above, we adapted the colors according to the new colors in the Orai1 models and we replotted the pore radius plot (Fig. 3) and hydration plots (Fig. 3,4). Otherwise for all time courses corresponding bar diagrams are shown for clarity.

ii. Describe the images in Supplemental fig A and B – make the Ca²⁺ binding pocket more clearly visible- difficult to differentiate between yellow and white.

The description is made clearer (Supplementary Fig. 5A,B) and the Ca²⁺ binding pockets (Fig. 4F,G) are visualized more clearly. The data are now in Fig. 4, according to your suggestion in iii (below).

iii. It might be useful to move Supple. Fig 3 A, B and J-L into main Figure 3. These data support the data in the main figure.

After intensive consideration and editing of our manuscript, we decided to keep Supplementary Fig. 3A,B in the Supplementary Information as similar cartoons are shown in Fig. 3. But we moved Supplementary Fig. 3J-L to Fig. 4 (Fig. 4F,G).

Figure 4 and later figures – please consider changing the colors on the models and making data clearly seen in the various experimental groups.

As described above, we adapted the colors in the models (Supplementary Fig. 6) and accordingly the colors in the rest of the figures.

Reviewer #2 (Remarks to the Author):

This paper by Hopf and colleagues aims to understand the mechanistic basis of gain-of-function (GOF) effects of a previously identified mutant, V181K (Tiffner et al, 2020, 2021). Introduction of some charged mutations at V181 (e.g., V181K/E/R) cause constitutive channel activity, and in the previous study, it was concluded that V181 is part of a network of “checkpoints” in Orai1 that controls channel gating. In the current study, the main advance here is addition of MD simulations showing that the water content around V181 increases with the V181K and V181E mutations and decreases with the loss-of-function V181F mutation. Additional analysis of the effects of mutating residues near V181 are also shown and inferences are made regarding putative functional interactions between V181 and these other residues based on alterations in current magnitudes with the additional mutations.

Although the manuscript has a lot of data and the idea of water in membrane proteins embedded within lipids is potentially interesting, I find the study to be mainly correlative, qualitative, and the evidence for increased water around the TM3/4 helices is based only on MD simulations with some critical theoretical assumptions. Further, although the potential biophysical mechanisms for why water increases around V181 with charged mutations is explained in the Discussion, the relevance of this phenomenon for gating/activation of the WT channel with the native Val remains unclear. Thus, the significance of the study appears more relevant to understanding the basis of the GOF phenotype of the mutant channel rather than improved understanding of the activation of the WT channel. In other words, what is the physiological relevance of water at the TM3/4 interface for the native channel?

We thank the reviewer for the constructive comments and positive evaluation of the manuscript.

We agree, that it is unclear whether water at the peripheral TM interfaces is critical for physiological pore opening. However, it is conceivable that STIM1-coupling to wt Orai1 may entail long-range C-terminal structural reorientations including TM4, possibly allowing water recruitment into the TM3/TM4-interface, where water would preferentially form hydrogen bonds with each other. Nevertheless, we agree that this requires further investigations in the future. Here, we demonstrate that water in peripheral TM interfaces tends to increase the area around V181K compared to V181. In support, we identified in additional experiments a correlation of constitutive current density levels and side chain size at/around V181. This dependence of pore opening and distance between TM3 and TM4/TM2' seems physiologically relevant. However, these findings we plan to publish in a separate manuscript.

Main points:

1) Figure 1E. It is important to show whether the various Orai1 V181X mutants bind to STIM1. Orai1+STIM1 currents don't change amplitude in most of the mutants. Is reduced current in some of the V181 mutants (in presence of STIM1) due to reduced STIM1 binding? or something else?

We tested for each group (charged, small polar, hydrophobic) several mutants for STIM1 binding. All of them show clear coupling to STIM1, in most cases to comparable levels like for wild-type, except Orai1-V181D (see Fig. S3). Hence, the inability of some mutants to not further enhance currents in the presence of STIM1, is less likely due to defect STIM1 coupling, except for Orai1-V181D.

2) Likewise, in Figure 2, a lack of change in current magnitudes in the presence of STIM1 (relative to without STIM1) suggests that STIM1 is not causing additional activation of the GOF Orai1 mutant. This is interesting and needs to be better understood. If water around V181 (in V181E for e.g.) is having a detrimental effect on STIM1-induced Orai gating, this suggests that the water at the TM3-4 interface may impede physiological (STIM1) activation of Orai1.

Analogously, we tested Orai1 A254K/E for coupling to STIM1. Interestingly, we found significantly reduced enhancement in FRET between Orai1 A254K/E and STIM1

(Supplementary Fig. 3E,F), which might be here an explanation for no significant increase in current of these mutants in the presence of STIM1.

3) P 9 and Figure S2F: Much is made of the change in FRET between the CFP and YFP tags at the C-terminal ends of Orai1. Specifically, a difference in FRET between the V181 mutants is interpreted as decrease in Orai1 “homomerization”. This interpretation is highly problematic for numerous reasons. The individual channels will have unknown and variable numbers of CFP- and YFP-tags that cannot be controlled within the channel population in each cell. The FRET measurement here is a vague and nebulous measure of some combined effect of conformation, orientation, and multimerization of the probes expressed at unknown ratios and the underlying phenomenon contributing to the FRET difference cannot be untangled. This data is not convincing and should be better substantiated or removed.

We agree and toned down our conclusion. Nevertheless, we decided to keep the data in the Supplementary Material, as the altered FRET provides an indication for structural changes due to mutation with amino acids with charged side chain. No conclusions are drawn regarding homo-/multimerization.

4) Figure 4. I am concerned about the quality of the simulations. Many of the helices in the Figures shown seem to be floppy, with broken helicity, some seem to be unravelling, and the channel seems to show a lot of asymmetry and is not well-behaved in these snapshot images. What does a movie look like? If the helices are unstable and unraveling, this could produce inaccurate metrics for water content which could lead incorrect conclusions. E.g., even in panel for the WT, the TM1 helix seems to be unravelling.

The apparent broken helicity or floppiness might come from the poor selection of the range of residues selected for the depiction of the configuration of the transmembrane region. Residues towards the end of the helices, which are outside the transmembrane region and belong either to a region that is more coiled-like after the selectivity filter, or the N-terminus from TM1, which protrudes towards the cytosol, indeed showed some unraveling toward the end. This is however expected as this region is more flexible and prone to unfold and refold during the simulation and common in many simulations. Snapshots depicting the protein shown from the side (Fig. 4; Supp. Figure 6) are provided to show that the integrity of the helices present in the transmembrane remains alpha helical. Additionally, we provide 2 videos (one top view, one side view) of Orai1 over 400ns simulation showing that the helices are not totally unfolded or destroyed.

5) There seems to be an unusually large amount of water within the TMs even in the WT channel, surprising for membrane proteins embedded in lipid. Obviously, this would depend on the simulation conditions, but a comparison to previous simulations of E190 mutations that group carried out in the Tiffner 2020 study might help provide some controls.

The same conditions have been used in the paper suggested by the reviewer. The presence of such high amount of water outside the conducting pore is indeed uncommon for a transmembrane protein and this is the reason why we chose to investigate the hydration present within the protein and outside the main pore itself as the mutations studied here occurred close to water-rich region yet outside the conducting pore. As a result, not many studies conducted on Orai had a look at the hydration within this region. The only MD study presenting results on this topic are the simulations from Alavizargar et al.¹ who studied the E190Q mutants, using a different protein model as ours, and linked the impaired selectivity of this mutant with the amount of water molecule present in the upper region of the channel. The snapshots provided by these authors also depict water-rich region around the main pore for the wild-type.

6) Figures 6. I understand the motivation for undertaking analysis of the double mutants at C143 and F253 with V181. The effects of these mutations are interpreted solely in the context of what they might do specifically to the GOF phenotype of the V181K mutant. However, it is equally likely that the reductions in V181K currents seen with hydrophobic substitutions are general in nature,

and also effect the activation/gating of unrelated Orai1 constitutively active Orai1 mutants such as H206A or V102X. This should be tested to provide stronger support for the proposed structural and functional interaction between C143/F253 with V181K.

As suggested, we investigated the effect of key mutations (C143A/W, F253A/W) on Orai1-V102A, -H134A and -F136S. All three mutants represent robust GoF mutants²⁻⁴. While F253A and C143A reduced or abolished Orai1-V181K currents, respectively (Fig. 6B-E), they did not or only slightly, but not significantly, reduce constitutive activity of Orai1-V102A, -H134A and -F136S (Supplementary Fig. 8I-N). In contrast, C143W and F253W, which increased the constitutive activity of Orai1-V181K (Fig. 6B-E), decreased the currents of Orai1-V102A and Orai1-H134A (Fig. 6I,J,L,M). Hence, the effects of C143- and F253-mutations cannot be generalized, and an inhibitory effect of C143A and F253A and an activating effect of C143W and F253W are likely specific to Orai1-V181K. This information is now included in the main text (p.13 l. 524-538).

7) Figure 7. Same issue as above. Are these generalized effects on channel gating independent of STIM1? Supplementary Figure 7G-H show that the A177X and F257X mutations inhibit currents of single mutants (even in the absence of V181K) by STIM1, suggesting that these are generalized effects on channel gating. Hence, the significance of these loss of these inhibitory mutations for the gating induced by V181K cannot be easily inferred.

Not all, but only some A177X and F257X mutants reduced Orai1 activation by STIM1 (Supplementary Fig. 9G-H). As suggested, we investigated the effect of key mutations (A177F/W, F257A/W, A177W F257W) on Orai1-V102A, -H134A and -F136S. While A177W, F257W and A177W F257W abolished constitutive activity of Orai1-V181K (Fig. 7B-E), they left constitutive activity of Orai1-V102A unaffected (Supplementary Fig. 9I,L). Orai1-H134A showed reduced, but not abolished constitutive activity upon A177W or F257W substitution (Supplementary Fig. 9J,M). Similarly, constitutive activity of Orai1 F136S remained unaffected by A177W, but was abolished by F257W (Supplementary Fig. 9K,N). F257A left constitutive activity of Orai1-V181K (Fig. 7D,E) as well as of Orai1-V102A and -H134A unaffected (Supplementary Fig. 9I,J,L,M). Hence, the effects of A177F/W, F257W and A177W F257W cannot be generalized, and distinctly affect Orai1-V181K and Orai1 GoF-mutants. This information is now included in the main text (p.13 l. 524-538).

Minor:

- Supplementary Figure 1E, I: The percentage of nuclear NFAT is shown as ~80% for V181G in panel E but is only about 30-40% in panel I. Please check and clarify.

Supplementary Fig. 1E only shows the maximum NFAT after 24h in 2mM Ca²⁺. Hence, we performed time course experiments and monitored the development of NFAT translocation over 4 hours upon the exchange from 0mM to 2mM Ca²⁺-containing solution (Supplementary Fig. 1E-J). Under these conditions NFAT translocation was reduced. Hence, the reason for the drastically increased NFAT translocation of the weakly constitutively active Orai1-V181G in Supplementary Fig. 1E is likely the prolonged exposure of the cells to Ca²⁺-containing media.

- Figure 2. The statistics are vague and exactly what the “*” mean is unclear. Which groups are being compared here?

The statistics are explained in the methods section, while the detailed F and p values are indicated in the Supplementary Table 2 in a separate file. In the diagrams we did not include all possible significance stars, but only the most important ones. The meaning of the shown significance stars is indicated in the legends.

- Figure 3. The amino acids should be clearly indicated at the various axial positions. It appears from the hydration plots that the primary areas of increase hydration are far into the TM1 inner segment, around residues R83 or even further away from the membrane. How is this relevant to channel gating and relief of the free energy barrier in the pore?

More pore-lining residues are now highlighted (Fig. 3). Concerning the relevance of the basic regions on pore opening, it has been recently reported that the inner basic pore region facilitates opening of the principal outer hydrophobic gate through a long-range effect involving hydration of the outer pore⁵.

References:

1. Alavizargar, A., Berti, C., Ejtehad, M. R. & Furini, S. Molecular Dynamics Simulations of Orai Reveal How the Third Transmembrane Segment Contributes to Hydration and Ca(2+) Selectivity in Calcium Release-Activated Calcium Channels. *J Phys Chem B* **122**, 4407–4417 (2018).
2. Derler, I. *et al.* The extended transmembrane Orai1 N-terminal (ETON) region combines binding interface and gate for Orai1 activation by STIM1. *J Biol Chem* **288**, 29025–29034 (2013).
3. Frischauf, I. *et al.* Transmembrane helix connectivity in Orai1 controls two gates for calcium-dependent transcription. *Sci Signal* **10**, (2017).
4. Butorac, C., Krizova, A. & Derler, I. Review: Structure and Activation Mechanisms of CRAC Channels. *Adv Exp Med Biol* **1131**, 547–604 (2020).
5. Yamashita, M. *et al.* The basic residues in the Orai1 channel inner pore promote opening of the outer hydrophobic gate. *J Gen Physiol* **152**, (2020).

Reviewer #3 (Remarks to the Author):

The manuscript by Hopf et al. used a combination of MD simulations and experimental assays to study the gating mechanism of Orai1 channels. This paper is well written and it represents a step forward in understanding the molecular basis of such mechanism. I can recommend the publication of the manuscript after the following points will be addressed:

We thank the reviewer for the constructive comments and positive evaluation of the manuscript.

1) The PDB code of the structure used for the computational part should be added in the first sentence of the MD simulations section to facilitate the readability of the text;

The PDB code is added in the first sentence of the MD simulation section. p. 6. l. 199

2) Which structure was used to produce the mutants? I suppose the structure of the wild type at the end of the dynamics but this should be clarified. Moreover, are the mutations introduced in all the subunits?

The mutations were introduced using the CHARMM-GUI web server and on the homology model directly rather than choosing a pre-equilibrated conformation of the wild type. The mutations were introduced in all the subunits. The requested information is included in the methods section. p. 6. l. 199

3) The number of the POPC molecules and the dimension of the simulation box should be provided;

Is there a specific reason why the authors used the TIP3P model which is different to that used in Frischauf et al? It is known from the literature that the water models can affect the results, especially if they play a key role in the study (see <https://pubs.acs.org/doi/10.1021/acsnano.0c04387>). Thus, the authors should comment on that;

The number of POPC molecules used for the membrane have been added to the methods section. The upper leaflet is made of 206 lipids while the lower leaflet is composed by 188 lipids with a box size initially 131.104 Å * 131.104 Å * 130.89 Å. p. 6. l. 200-202

The water models are of course important and can strongly influence the outcome of a simulation as the reviewer kindly reminded us. The TIP3P model was used as it is the common water model to use together with the CHARMM force field. The OPLS forcefield was used in the Frischauf et al.¹ study.

4) The authors should specify what method or program they used to predict the protonation state of the residues;

The protonation states of the residues are set to pH=7 except for the glutamic acid in the V181E where both states were investigated.

5) What are the trajectories used to compute the RMSD profiles shown in Fig. S3? Is a plateau reached in all the replicas?

For each system (wild type and mutants), the three replicas were used to compute the RMSD. After extension of the MD simulations to 400ns all mutants, except Orai1-V181E, reached a plateau. Further extension of Orai1-V181E simulation to 550 ns led to a development of a plateau. This is stated in the methods section and legends. p. 6. l. 205

6) What are the structures used to compute the radius profiles with the HOLE program? Did they

refer to the profile computed on the structures at the end of their dynamics or are they averaged over the 50 ns used for the post-processing analyses?

The structures used to compute the radius profiles are extracted from the last 50ns of each replicas. The final HOLE profile is thus an average of all the radius profiles from the configuration present in the last 50ns (between 350 and 400ns for all mutants, except Orai1-V181E where analysis was done between 500 and 550ns).

7) In the same vein, is the number of water molecules computed in a single frame at the end of the dynamics for the three replicas or do the profiles shown in Fig. 4C refer to the mean value over the overall last 50 ns and then averaged for the three replicas?

The same treatment was done for the water molecules as for the radius profile, it represents the mean value over the last 50ns averaged over three replicas.

8) Page 10: when the authors state “In contrast, the mutants did not show the same level of structural stability, with RMSD values still increasing at 250 ns”, do they believe that these results are reliable even if the mutants did not reach their stability? Wouldn't it be more prudent to extend these simulations, especially in the systems V181F where the curves of TM3 and TM4 are still increasing?

Admittedly, the RMSDs did not reach a plateau or only just in certain cases notably showing that the protein did not reach an equilibrated state. We thus extended the simulations in order to improve the reliability of the calculations. By extending the simulation time up to 400ns most of the cases studied here now present stabilized RMSDs. The only ill-case was V181E in the deprotonated form where extending the trajectories up to 550 ns was needed.

9) It's not clear the statement “A deeper investigation of the water crevice around V181 revealed the involvement of residues C143 in TM2, A177 in TM3 and F253 in TM4”. Did the authors do a quantitative analysis to characterize their role or did they observe only the structure to deduce that? One can argue that also neighboring residues can play a role in that.

It is unfortunately more of a qualitative analysis as no simulations have been carried out to investigate the effect of these residues on the local hydration. Hence, we changed the sentence to: A closer look at the surrounding of V181 and K181 indicated that especially the residues C143 in TM2, A177 in TM3 and F253 in TM4 are directed to the water crevice around position 181 (Figure 4E). p. 12, l. 450-453

10) Page 11: the sentence “...the area formed by these three positions tended to increase concomitantly with the amount of water (Figure S3K,L)” seems to be slightly an overstatement if the error bars in panel L are considered.

We agree that the sentence is an overstatement considering the overlapping error bars. Meanwhile, we extended the trajectories and as for the rest of the manuscript, the values (means and standard deviations) used for this plot is from an average over the areas obtained from the three different replicas. The average area value in each replica was computed over all six subunits over the last 50 ns. We see a clear, although not significant difference in the area, in particular for V181K and V181E. Hence, we rewrote the sentence the following way: In particular, for the mutants Orai1-V181K and Orai1-V181E, the area formed by these three positions tended to increase. p. 12, l. 450-453

11) Again in Fig. 5 (and also in S5) it's not clear which data have been used to compute the angles and the distances. Do they refer to a single subunit? Are they computed on a single trajectory or on all the replicas? Are they computed at the end of the dynamics or in the last 50 ns?

It is an average over all subunits in all the replicas and the last 50 ns were used.

12) Page 14: the authors should provide a profile of the interaction R181-F257 during the

simulations to state “In our simulations, this difference was evident in the ability of R181 to form a cation- π interaction with the phenyl ring of F257.”

An excellent suggestion, the distances between the heavy atoms in the guanidinium moiety of R181 and the heavy atoms in the phenyl ring of the closest F257 have been calculated (analogously to the previous results presented in this manuscript, the last 50 ns of the trajectories were used, all the six subunits in all three replicas were used to compute the distribution profile of the distances). We also added the distribution of the distances between the residues K181 (the nitrogen from the amino moiety was here chosen) and F257 in the trajectories from the V181K mutants in order to give a better appreciation for the strength of the interactions between R181 and F257 (Supp Fig. 7G).

Moreover, we calculated the angle between the normal of the phenyl ring and the normal to the guanidinium. In the scatter plot below (Fig. 1) the angles are shown as a function of the distances. Hence, at lower values of distance when the two moieties are closer, the angle values are more spread towards 0 or 180 (the normal vectors are parallel or antiparallel to each other showing that some stacking is indeed happening). At higher distance values, the angles values become more random since the stacking is not there. The orientations in the MD are compatible with density functional theory (see Fig 4 in Gallivan et. al.² for preferential orientations of R side chain interacting with W side chain arising from density functional theory (Hartree Fock level of theory with 6-31G basis set)), but in our case cation- π stacking is imposed by steric restraints arising from the local tertiary structure of the Orai1 complex. The obtained angles do not follow from our force field model.**

Although cation- π interaction is happening, it is not evidently systematic and more nuanced. Notably, orientational aspects of cation- π interactions are not explicitly modeled in the devised CHARMM36 force field, rationalizing this result. Future research may include an improved cation- π description using an NBFIX optimization to CHARMM36, as introduced in Liu et al.³.

Figure 1: Distribution of the angles between the normal of the phenyl ring (F257) and the normal to the guanidinium (R181). Left: The scheme shows vectors used to calculate the angles. The vector normal to the phenyl ring is a cross-product between two vectors in the plane of the phenyl ring. One vector defined by the center of mass (COM) of the phenyl ring and the carbon ξ and another vector is defined by the COM of the phenyl ring and the ϵ 1 carbon. The vector normal to the guanidinium is a cross-product between two vectors in the plane of the guanidinium moiety. One vector is defined by the guanidinium carbon and one of the ending nitrogen atom and another vector defined by the guanidinium carbon and the other ending nitrogen atom. **Right:** Scatter plot showing the angles between the normal of the phenyl ring and the normal to the guanidinium as a function of the distance

between the heavy atoms in the guanidinium moiety of R181 and the heavy atoms in the phenyl ring of the closest F257.

13) Can the authors comment on their results in the view of <https://doi.org/10.3389/fmolb.2021.773388?>

We cited this manuscript in the discussion section. p. 15, l. 590-593

References:

1. Frischauf, I. *et al.* Transmembrane helix connectivity in Orai1 controls two gates for calcium-dependent transcription. *Sci Signal* **10**, (2017).
2. Gallivan, J. P. & Dougherty, D. A. Cation- π interactions in structural biology. *Proceedings of the National Academy of Sciences* **96**, 9459–9464 (1999).
3. Liu, H., Fu, H., Shao, X., Cai, W. & Chipot, C. Accurate Description of Cation- π Interactions in Proteins with a Nonpolarizable Force Field at No Additional Cost. *J Chem Theory Comput* **16**, 6397–6407 (2020).

Reviewer #2 (Remarks to the Author):

The revised manuscript addresses some of the concerns raised in the previous round, but continues to suffer from major concerning problems of interpretation and claims that are not supported by the results.

We thank the reviewer for the constructive comments.

The paper is also very difficult to understand and in many cases, the results presented are not interpreted at all.

The abstract concludes with a bold statement that “We propose a gating mechanism involving widening of peripheral TM-interfaces which may facilitate enhanced hydration after STIM1 binding to trigger Orai1 opening”. There is no data in the paper supporting this sweeping statement. On the contrary, the data in Figure 1E clearly show that the mutants (E, R, Q, K), that evoke water at the TM3-TM4 interface are unresponsiveness to STIM1 binding. The D mutant in fact does not even bind STIM1 adequately (Supplem Figure 3). This statement is not supported by the data shown and must be changed.

We apologize for the exaggeration with this sweeping statement. We agree that we did not show that the binding of STIM1 triggers a broadening and hydration of peripheral TM interfaces.

It is true that Figure 1E clearly shows that the mutants (E, R, Q, K), that evoke water at the TM3-TM4 interface are low responsive or unresponsiveness to STIM1 binding. The D mutant is not active in absence and only weakly active in the presence of STIM1, thus indicating a distinct effect. For these effects we provide a more detailed discussion in the new revision of the manuscript (p. 15/16, l. 575-589).

Nevertheless, our data might give a hint to a possible STIM1-induced Orai1 activation mechanism. Therefore, we have modified our last statement in the abstract: “*We conclude that a positive charge close to the center of TM3 or TM4 facilitates concomitant hydration and widening of peripheral TM interfaces to trigger constitutive Orai1 pore opening to a level comparable to or exceeding that of native activated Orai1.*” p. 2, l. 30-32

Moreover, in the discussion we highlight again the process of activation by charged substitutions ‘possibly mimics the physiological pore opening mechanism triggered by STIM1’ (p. 15, l. 573-574). We are not saying that we showed that STIM1 induces widening and hydration of the peripheral TM interfaces.

The abstract also makes the claim that the MD simulations show that the V181X gain-of-function mutants show local swelling at the channel periphery. I do not see a correlation between the “local swelling”, channel activity, and water anywhere. Crucially, the F mutant which shows more area between the 143, 177 and 253 is not even active (and hence negates the point of this abstract statement). The study should carry out analysis for the other mutants such as Y, L or S mutants which are active with STIM1 (but not w/o STIM1) to support the central conclusion of the paper.

The abstract states: “Focusing on the gating mechanism operating along the peripheral transmembrane domain (TM) 3/TM4-interface, we report here that some charged substitutions close to the center of TM3 or TM4 lead to constitutively active Orai1 variants triggering nuclear factor of activated T-cell (NFAT) translocation into the nucleus. Molecular dynamics simulations unveil that this gain-of-function correlates with enhanced hydration at peripheral TM-interfaces, leading to increased local structural flexibility of the channel periphery and global conformational changes permitting pore opening.” Thus, in the abstract we specifically connect only gain-of-function due to charged substitutions with enhanced hydration.

We added ‘some’ to indicate that it holds for K,R,E-substitutions, but not for the D. Orai1-V181D remains inactive in the absence and is only weakly active in the presence of STIM1, possibly due to decreased side chain size compared to E, as discussed now on p. 15/16, l. 575-589.

We are aware of the fact that also V181F leads to an enhanced area formed by C143 in TM2, A177 in TM3 and F253 in TM4, which is probably due to the bulkiness of the F, but not due to increased hydration. The loss-of-function (LoF) of Orai1-V181F could arise from a water shielding effect arising from the phenylalanine side chain maintaining a dewetted TM3/TM4-interface. Comparing Orai1-V181F with wild-type (wt) reveals an improved water shielding because of hampered activation not only in the absence, but also in the presence of STIM1. This stimulated our ongoing research. Early results indicate a potentially more prominent role of hydrophobic interactions in the TM3/TM4 interface. On the one hand we found that the exceptional LoF of Orai1 V181F results from inhibitory intra- and inter-subunit interactions specific to a phenylalanine at this position. On the other hand, other larger aromatic amino acids or even double substitutions with aromatic amino acids led to constitutive activity, indicating that bulky amino acids close to the center of TM3 and/or TM4 can also lead to gain-of-function (GoF). Based on these results, we believe that pore opening can occur due to increased hydration followed by increased local structural flexibility as reflected by increased RMSD values (Supp. Fig 5), or simply due to bulky amino acids causing dilation. However, we believe that these new findings are way beyond the scope of this manuscript and prefer to include them in a separate manuscript. Nevertheless, we have devoted a separate paragraph to the Orai1 V181F case in the discussion (p. 16, l. 616 – 627; p. 17, l. 661 - 662).

Although the paper contains a large amount of data, much of it is very difficult to follow and in most cases, the crucial pieces of information are not even interpreted by the authors. For example, what does it mean that STIM1 does not activate the V181 gain-of-function mutants? Or that in some cases (D, E), they do not effectively bind to STIM1? The lack of gating by STIM1 is simply ignored.

We agree with the referee that it is important to discuss all the results and we added now an interpretation of the functional STIM experiments in the discussion section. p. 15/16, l. 575-589

Our functional screen of Orai1-V181X mutants revealed that in particular some charged substitutions K, E and R led to robust constitutive activity in the absence of STIM1. Interestingly, the V181D-substitution left the channel inactive, possibly due to decreased side chain size compared to E. Investigations in the presence of STIM1 revealed weak (Orai1-V181K) or no further current enhancements (Orai1-V181E/R). Orai1-V181D showed significantly reduced STIM1-induced activation compared to wt Orai1. The reduced or lacking responsiveness of these mutants with charged substitutions to STIM1 may underlie their altered coupling. Indeed, V181E/R-substitution in Orai1 led to delayed and slightly reduced and V181D-mutation resulted in significantly reduced FRET with STIM1, while FRET of Orai1-V181K with STIM1 exhibited comparable behavior like wt Orai1.

Nevertheless, Orai1-V181K/E/R showed reduced homomerization FRET, potentially due to changed conformational rearrangement of the Orai1 C-termini, which might be another reason why Orai1-V181K/E/R-mutants are not further activated by STIM1.

Concerning the reduced excitability of Orai1 V181D, we presently speculate that this observation is driven by altered interaction partners within Orai1 V181D and in particular its C-terminus which features numerous positively charged moieties (Orai1 C-terminus: aa266-301: TDRQFQELNELAEFARLQDQL **DHRGDH**PLTP GSHYA (note that histidines neighbouring acidic residues (D) are very likely to be protonated at physiological pH); the residues relevant for STIM1 binding are aa267-286, after the STIM1 binding site are clusters of charges (highlighted in red), which could interfere with Orai1 V181D function). This possibility will be studied in our future investigations.

Nevertheless, the main conclusion of the paper results already from the data without STIM1. A charge close to the center of TM3 or TM4 can induce CRAC channel-like activity already in the absence of STIM1. We used this finding as a motivation to get an idea about the opening mechanism induced by these charged side chains.

One more crucial point: A careful examination of the current amplitudes in the absence (or even in the presence!) of STIM1 shown in Figure 1E and the amount of water in the TM3-TM4 interface shown in Figure 4D shows basically no correlation between the amount of water and the current magnitude. For example, the E mutation shows the maximal water, but this mutant is considerably less active than the K mutant which shows much less current. The F mutation shows some degree of water but is associated with virtually no current. The amount of water in the Ala mutant is about the same or less than the WT protein but this mutation does show some constitutive current. Basically, I do not see the functional correlation between the amount of water in the TM3-TM4 interface and channel activity. This undercuts the major premise and key conclusion of the paper.

To investigate for a potential correlation of current (in the absence of STIM1) with pore radius (with 0 Å, 20 Å, 30 Å distance to the selectivity filter), pore hydration (with 0 Å, 20 Å, 30 Å distance to the selectivity filter) or peripheral hydration (with 20 Å distance to the selectivity filter, as in the plane of V181), we investigated for their linear correlation (Figure 1) and estimated the Pearson Correlation coefficient, a measure of linear correlation (Table 1). Most of them showed a Pearson correlation coefficient in the range of 0,6 – 0,8, indicating a moderate to strong correlation (see Table 1). The only exception is the reduced correlation of current with pore hydration with 0 Å distance to the selectivity filter (Pearson correlation coefficient: 0,3). Overall, our findings indicate a clear correlation of pore radius or hydration and current size, in particular along the basic region. We did not investigate for a correlation of radius or hydration with currents in the presence of STIM1, as MD simulations were only performed in the absence of STIM1.

The reason for the higher amount of water in the pore and the periphery of Orai1 V181E is discussed in the results (p. 13, l. 450-453). “Interestingly, Orai1-V181E exhibited even higher back pore hydration. This more pronounced hydration for V181E could possibly be explained by the relatively small size of the glutamic acid side chain compared to arginine and lysine. The latter two possibly occupy the positions that would be occupied by water molecules in the case of glutamic acid.”

The small differences in the peripheral hydration of Orai1-V181A and Orai1-V181F are in the range of the standard deviation. From our studies presented in this manuscript we can clearly conclude that some charged amino acids on V181 increase the peripheral hydration and thus also the pore

hydration. This provides a first idea of a possible pore opening mechanism. Concerning Orai1-V181A it can be concluded that reduced peripheral hydration leads to reduced pore hydration. However, whether a slight increase in peripheral hydration in Orai1-V181A compared to wt Orai1 is required for pore hydration remains unclear. A slight increase in hydration at the TM3/4 interface in Orai1-V181F could probably be compensated by strong hydrophobic shielding, thus leading to loss of function. Hence, there also appears to be a clear influence of hydrophobic interactions in peripheral TM interfaces, which probably control pore opening. Our initial studies on aromatic double substitutions around V181 show that pore opening can also occur despite hydrophobic shielding, probably solely due to large side chain.

Summarizing, peripheral hydration is one possible reason for pore opening, which occurs in Orai1 mutants containing charged substitutions at V181. This aspect is fully covered by the present story. Alternatively, pore opening could probably also be induced by dilation of peripheral TM interfaces which may be additionally controlled by hydrophobic interactions. However, as said, the latter aspects are far beyond the scope of this study and will be addressed in detail in a future manuscript.

Figure 1: Linear correlation of pore radius, pore hydration or peripheral hydration versus currents Orai1 V181K/E/R/A/F compared to wt Orai1. a) – g) Dot diagrams showing pore radius with 0, 20 and 30 Å to the selectivity filter (a-c), pore hydration with 0, 20 and 30 Å to the selectivity filter (d-f) and peripheral hydration with 20 Å to the selectivity filter (g) over current of Orai1K/E/R/A/F versus Orai1 and their linear fit.

Table 1: Pearson correlation coefficient of pore radius, pore hydration or peripheral hydration versus currents Orai1 V181K/E/R/A/F compared to wt Orai1.

Distance to selectivity filter	Pearson correlation coefficient of current versus		
	pore radius	pore hydration	periphery hydration
0 Å	0,66	0,30	

20 Å	0,84	0,62	0,72
30 Å	0,76	0,68	

Finally, the positions of the TM1 residues along the y axis needs to be indicated in Figure 4C. This was requested in the initial review as well. This is essential for understanding the relative amounts of water along the pore.

This is now adapted accordingly.